# High thermoelectric performance enabled by convergence of nested conduction bands in Pb$_7$Bi$_4$Se$_{13}$ with low thermal conductivity

Lei Hu [1,2✉], Yue-Wen Fang [2], Feiyu Qin[2], Xun Cao [1], Xiaoxu Zhao[1], Yubo Luo[1], Durga Venkata Maheswar Repaka [3], Wenbo Luo[4], Ady Suwardi [3], Thomas Soldi [5], Umut Aydemir[6,7], Yizhong Huang[1], Zheng Liu [1], Kedar Hippalgaonkar[1,3], G. Jeffrey Snyder[5], Jianwei Xu [3] & Qingyu Yan [1✉]

Thermoelectrics enable waste heat recovery, holding promises in relieving energy and environmental crisis. Lillianite materials have been long-term ignored due to low thermoelectric efficiency. Herein we report the discovery of superior thermoelectric performance in Pb$_7$Bi$_4$Se$_{13}$ based lillianites, with a peak figure of merit, $zT$ of 1.35 at 800 K and a high average $zT$ of 0.92 (450–800 K). A unique quality factor is established to predict and evaluate thermoelectric performances. It considers both band nonparabolicity and band gaps, commonly negligible in conventional quality factors. Such appealing performance is attributed to the convergence of effectively nested conduction bands, providing a high number of valley degeneracy, and a low thermal conductivity, stemming from large lattice anharmonicity, low-frequency localized Einstein modes and the coexistence of high-density moiré fringes and nanoscale defects. This work rekindles the vision that Pb$_7$Bi$_4$Se$_{13}$ based lillianites are promising candidates for highly efficient thermoelectric energy conversion.

[1] School of Materials Science and Engineering, Nanyang Technological University, Singapore, Singapore. [2] Materials and Structures Laboratory, Tokyo Institute of Technology, Yokohama, Japan. [3] Institute of Materials Research and Engineering, A*STAR (Agency for Science, Technology and Research), Singapore, Singapore. [4] Institute for Advanced Materials, North China Electric Power University, Beijing, China. [5] Department of Materials and Science Engineering, Northwestern University, Evanston, IL, USA. [6] Department of Chemistry, Koc University, Sariyer, Istanbul, Turkey. [7] Koc University Boron and Advanced Materials Application and Research Center, Sariyer, Istanbul, Turkey. ✉email: leihuedu@gmail.com; AlexYan@ntu.edu.sg

Worldwide energy and environmental crisis have an urgent requirement on clean and sustainable energy sources due to the combustion of fossil fuels. Thermoelectric (TE) semiconductors provide promising opportunities in enabling the conversion of waste heat into electrical energy, especially favorable for relieving the current dilemma[1–4]. The crucial issue is the relatively limited energy conversion efficiency, related to the TE figure of merit, $zT = S^2\sigma T/(\kappa_e + \kappa_L)$. Here, $S$ and $\sigma$ are Seebeck coefficient and electrical conductivity; $T$ is absolute temperature; $\kappa_e$ and $\kappa_L$ are electronic and lattice thermal conductivities, respectively. Electronically, the intertwining of $S$, $\sigma$, and $\kappa_e$ complicates efforts to optimize TE performance. With the single parabolic band assumption, the $zT$ can be described as $zT = zT(\eta, B)$ by the conventional quality factor analysis[5,6]. This description correlates $zT$ solely with two independent physical parameters, i.e., tunable reduced Fermi level, $\eta$ and quality factor, $B$. This $B$ value is an effective descriptor with considering band degeneracy ($N_v$), inertial effective mass ($m_I^*$), deformational potential ($\Xi$), and lattice thermal conductivity, according to the relationship of $B \sim N_v/(m_I^* \Xi^2 \kappa_L)$[6].

The optimization of electrical properties and simultaneous suppression of heat transport are essential and critical for advanced TE materials. To achieve the optimal electrical properties, effective strategies on electronic band engineering are indispensable, such as electronic band convergence[7–9], resonant level introduction[10,11], band inversion and flattening[12,13]. Meanwhile, low lattice thermal conductivity is equivalently favorable. Intrinsically, the reduced heat transport requires large lattice anharmonicity[14,15], complex crystal structures[16], localized Einstein modes[17,18], and suppressed transverse acoustic phonon branches[19,20]. Extrinsically, lattice imperfection engineering, including point defect[21,22], dislocation[23,24], grain boundary and interface[25], coherent nanostructure[26,27], etc., is equally prevalent to scatter all-length scale phonons and thus impede heat propagation.

The alignment of multiple electronic bands and intrinsically low lattice thermal conductivity are highly beneficial for promising TE materials. Ternary heavy metal chalcogenides have emerged as potential candidates for optimal TE materials, especially the $Pb_{N-1}Bi_2Se_{N+2}$ lillianite homologous series[28–30]. These $Pb_{N-1}Bi_2Se_{N+2}$ homologous series comprise of NaCl-type layered modules with different edge-sharing octahedra. Historically, lillianite homologs have been overlooked due to their intrinsically inferior electrical properties. Theoretically, by systematically modulating the building blocks of $(PbSe)_m$ and $(Bi_2Se_3)_n$, intriguing electronic properties could be still anticipated. Furthermore, heavy constituent elements, complex crystal structure, and lone pair $6s^2$ electrons[31] in $Pb^{2+}$ and $Bi^{3+}$ collectively give rise to limited phonon propagation, large lattice anharmonicity, and thus low lattice thermal conductivity. However, it is still a grand challenge to discovery promising TE performance in lillianite homologs.

Here, we report an unconventional TE compound, $Pb_7Bi_4Se_{13}$. By a series of heterogeneous element doping (Ga, In, Ag, and I), a high maximum $zT$ of 1.35 at 800 K and a decent average $zT$ of 0.92 from 450 to 800 K are concurrently realized for $n$-type $(Pb_{0.95}Ga_{0.05})_7Bi_4Se_{13}$, which becomes the highest performance realized not only in lillianites but also in a series of compounds with similar structures. This optimal TE performance electronically stems from the alignment of nested conduction bands. A distinctive quality factor is delicately set up to screen and evaluate TE performance. This developed quality factor incorporates nonparabolic bands and bipolar effect. It breaks the limitation of the conventional quality factor that uses single parabolic band assumption. By adopting this, a more promising TE performance is also accessible with enhanced quality factors. Equally significantly, the lattice thermal conductivity is exceptionally low of 0.17 W m$^{-1}$ K$^{-1}$ at 800 K, which originates from large lattice anharmonicity, low-frequency localized Einstein modes, and the coexistence of high-density moiré fringes and nanoscale defects, especially periodic defective stripes with high-density stacking faults and intense lattice strains. Our work demonstrates that $Pb_7Bi_4Se_{13}$ based lillianites hold a great potential to be efficient TEs. Importantly, this work also develops an unconventional approach to calculate quality factors, highly favorable for accelerating the screen and evaluation of advanced TE systems.

## Results

**TE performance.** $Pb_7Bi_4Se_{13}$ crystallizes in a monoclinic structure with the space group (No. 12) of $C2/m$[28]. This monoclinic structure consists of diverse polyhedra, including $PbSe_6$, $PbSe_7$, and $BiSe_6$ (Fig. S1a). The synthesis of $Pb_7Bi_4Se_{13}$ includes ball-milling and long-term annealing with details presented in the experimental section. A series of heterogeneous elemental doping has been carried out by introducing Ga, In, Ag, and I, according to the nominal compositions of $(Pb_{1-x}Ga_x)_7Bi_4Se_{13}$ ($x = 0$, 0.02, 0.05, and 0.1), $(Pb_{1-x}In_x)_7Bi_4Se_{13}$ ($x = 0.05$ and 0.1), $(Pb_{1-x}Ag_x)_7Bi_4Se_{13}$ ($x = 0.05$ and 0.1) and $Pb_7Bi_4Se_{13-x}I_x$ ($x = 0.4$ and 0.8). The XRD patterns of pristine and doped $Pb_7Bi_4Se_{13}$ are presented in Fig. S1b. This distinct doping achieves varied electron concentrations at 300 K, ranging from $1.0 \times 10^{20}$ to $1.2 \times 10^{21}$ cm$^{-3}$. For the convenience of comparison and discussion, $Pb_7Bi_4Se_{13}$ with different element doping are denoted as the corresponding electron concentration ($n_H$) measured at 300 K, as shown in the inset of Fig. 1. The composition, carrier concentration, Hall mobility, and geometry density are tabulated in Table S1. It should be noted that there exists a certain degree of anisotropy of TE properties, due to the monoclinic structure of $Pb_7Bi_4Se_{13}$. The measurement geometry and TE properties parallel and perpendicular to the spark plasma sintering (SPS) directions of two different samples are presented in Fig. S3. In this work, only the TE properties parallel to the SPS direction is presented and discussed for convenience. It should be noted that $zT$ value calculations are consistent from the direction of electrical conductivity, Seebeck coefficient, and thermal conductivity. For instance, $(Pb_{0.95}Ga_{0.05})_7Bi_4Se_{13}$ corresponds to the sample with $n_H$ of $1.2 \times 10^{20}$ cm$^{-3}$ (also denoted as Ga2). As shown in Fig. 1a, the electrical conductivities, $\sigma$ of $Pb_7Bi_4Se_{13}$ alloys with $n_H$ larger than $1.6 \times 10^{20}$ cm$^{-3}$, show a decreasing tendency with increasing temperature, which is a characteristic of degenerated semiconductors. The $\sigma$ of $Pb_7Bi_4Se_{13}$ alloys with lower $n_H$ decreases firstly and then increases slightly with rising temperature. Specifically, $\sigma$ of $Pb_7Bi_4Se_{13}$ with $n_H$ of $1.2 \times 10^{20}$ cm$^{-3}$ decreases from 250 Scm$^{-1}$ at 300 K to 104 Scm$^{-1}$ at 723 K, followed by a slight increase to 119 Scm$^{-1}$ at 800 K. The magnitude of Seebeck coefficient, $|S|$, of $Pb_7Bi_4Se_{13}$ with $n_H$ higher than $2.8 \times 10^{20}$ cm$^{-3}$ increases monotonously with increasing temperature. And the $|S|$ of samples with lower $n_H$ demonstrates broad extrema at high temperatures, attributed to the contribution from the minority carrier. Taking the sample with $n_H$ of $1.2 \times 10^{20}$ cm$^{-3}$ as an example, the $S$ at 300 K is $-89$ $\mu$VK$^{-1}$ and decreases to $-217$ $\mu$VK$^{-1}$ at 723 K and then increase to $-213$ $\mu$VK$^{-1}$ at 800 K. According to Goldsmith–Sharp bandgap, $E_g = 2e|S_{max}|T_{max}$, the $|S_{max}|$ and $T_{max}$ are the maximum of the magnitude of Seebeck coefficient, and its corresponding temperature, respectively. The bandgap is estimated to be about 0.33 eV, close to the reported value of 0.29 eV[28]. The total thermal conductivities of $Pb_7Bi_4Se_{13}$ demonstrate low values at 300 K, e.g., 0.48 Wm$^{-1}$ K$^{-1}$ in $Pb_7Bi_4Se_{13}$ with $n_H$ of $1.2 \times 10^{20}$ cm$^{-3}$, which further decreases to 0.32 W m$^{-1}$ K$^{-1}$ at 800 K. By systematic doping, a series of $zT$ is achieved. And the peak $zT$ of 1.35 is obtained at 800 K for the composition of $(Pb_{0.95}Ga_{0.05})_7Bi_4Se_{13}$ with $n_H$ of $1.2 \times 10^{20}$ cm$^{-3}$

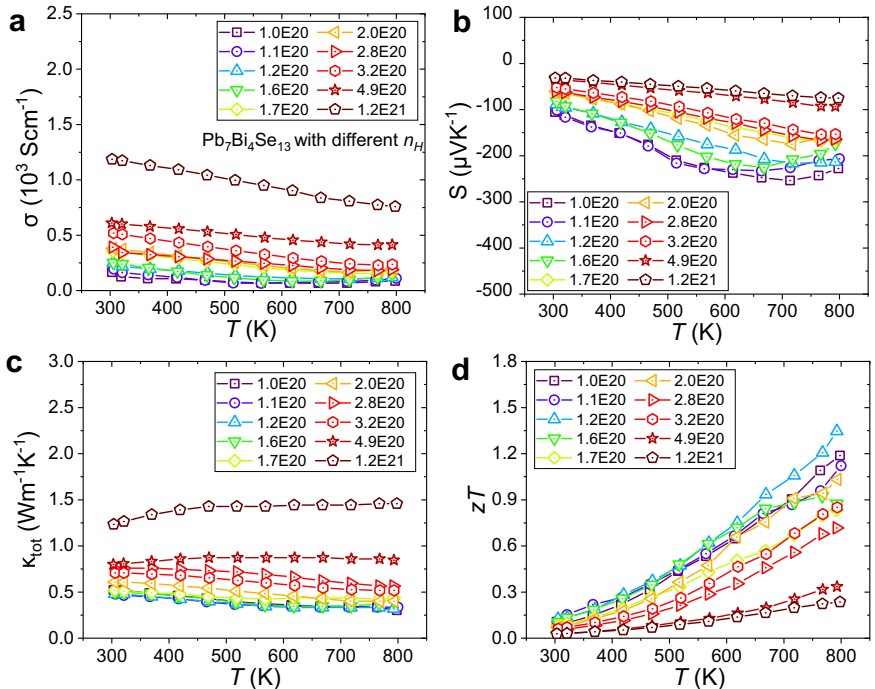

**Fig. 1 Thermoelectric properties. a** Electrical conductivity, **b** Seebeck coefficient, **c** total thermal conductivity, and **d** dimensionless figure of merit, $zT$.

(Fig. 1d). It is worth noting that this is the highest $zT$ achieved to date not only in lillianite homologous series, but also in a series of compounds with similar structures, such as Tetradymite, Pavonite, Cannizzarite, Galenobismuthite, complex rare-earth sulfides, and other structure-similar compounds. The total number of 37 different compounds was enumerated and compared in Fig. S2, which shows the competitive performance of $Pb_7Bi_4Se_{13}$. The detailed lattice thermal conductivities and $zT$ values of 37 distinct compounds are tabulated in Table S2. Compared to the state-of-the-art TE materials, like PbTe[7], SnSe[32], GeTe[33,34], and $Mg_3Sb_2$[35], etc., this result demonstrates a great potential for realizing high TE performance in lillianite-type or similar structures. These promising high peak and average $zT$ values enable $Pb_7Bi_4Se_{13}$ to be applicable for middle-temperature power generation, such as waste energy recovery, remote sensor power, and emergency power sources. Besides, it also could be employed in the marine engines of ships by utilizing waste heat from exhaust pipes. Furthermore, thermomechanical properties are also investigated in $Pb_7Bi_4Se_{13}$. The compression test shows an ultimate compressive strength of 85 MPa and the strain can reach 1.2% (Fig. S6). Nanoindentation investigation indicates the hardness of 2.88 ± 0.04 GPa (Fig. S7). And the microstructural investigation shows no obvious precipitates and micropores in the sample after annealing under vacuum conditions for 2 weeks (Fig. S8), which shows its reasonably robust thermal stability. The heating and cooling measurements is also presented in Fig. S5.

**Convergence of nested conduction bands**. To understand the origin of such high TE performance, we performed DFT calculations to study the electronic structure of $Pb_7Bi_4Se_{13}$. Figure 2a shows the band structure along with the high symmetry **k**-point path. The conduction bands at $M_2$ and $Y_2$ points demonstrate large $E \sim$ **k** dispersion, implying the light band characteristic. The feature is the band nestification especially occurred at $M_2$ and $Y_2$. These bands have nearly the same energies, i.e., the difference in the conduction band minimum (CBM) at $M_2$ point is around 100 meV, which can synergistically participate in the electrical transport. More significantly, the effectively nested conduction

bands at $M_2$ and $Y_2$ also demonstrate nearly indiscernible energy separations. In details, the energy difference of CBMs of $M_2$ and $Y_2$ is as small as 8 meV. In this way, the nested conduction bands at $M_2$ and $Y_2$ reveal a high number of valley degeneracy, comparable to many decent TE materials, such as n-type Si[1], $Bi_2Se_3$[36], and PbTe[7], as well as p-type elemental Te[37]. The convergence of nested bands enables multiple conducting channels without the deterioration on the Seebeck coefficient, highly favorable for superior electrical performances. A recent study suggests the convergence of electronic bands at distant **k** points is superior[38]. However, for semiconductors with lower symmetries, this favorable band configuration is a grand challenge. Alternatively, the convergence of nested bands could be significant to advance electrical properties for lower-symmetry TE materials. The convergence of nested bands plays an important role in charge transport properties than the situation with only heavy bands, even though it introduces a certain degree of intervalley scattering. This situation is confirmed by the theoretical calculation in the recent work[38], in which the power factors in band configuration of nested bands are still higher than that in band configuration of the heavy band only. Experimentally, the band nestification is proven to be an advanced method in the Tellurium TE compound with space for improvement[38]. Furthermore, the convergence of nested bands holds great potential for TE materials, if nested bands could be removed away from the same **k** point by chemical modifications.

In regard to valence band structure, the valence band maximum (VBM) exhibits flatten dispersion along $\Gamma$-$Y_2$ and $\Gamma$-$M_2$-D lines, which is a characteristic of heavy valence valleys. The VBM lies in the middle of $\Gamma$-$Y_2$ and $\Gamma$-$M_2$ paths, implying high valley degeneracy. Therefore, the multiband feature in both conduction and valence valleys enables $Pb_7Bi_4Se_{13}$ alloys to be promising for high TE efficiency. The spin–orbital coupling (SOC) effect on the band structure is also considered. The introduction of SOC has an indiscernible effect on the band dispersions of CBM and VBM, as shown in Fig. S9. The calculated $E_g$ is suppressed from 0.67 eV (without SOC) to 0.22 eV (with SOC). Figure 2b exhibits the orbital projected density of states

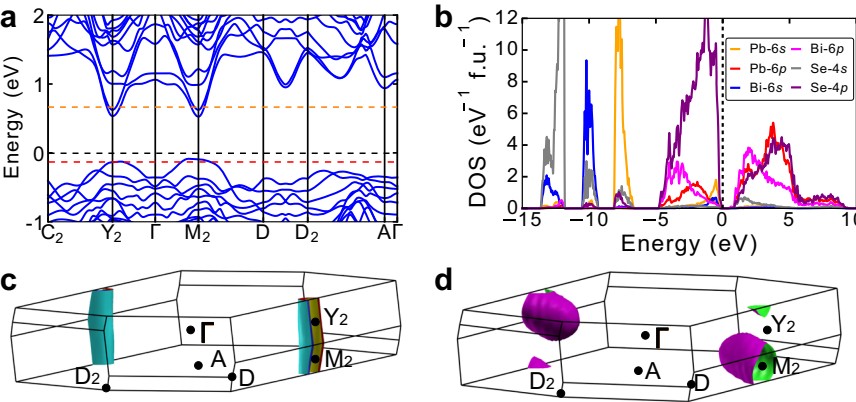

**Fig. 2 Electronic structure and Fermi surfaces. a** Band structure. The dashed black line indicates the Fermi level of pristine $Pb_7Bi_4Se_{13}$. The orange and red dashed lines correspond to the Fermi levels of 0.05 electron- and 0.05 hole-doped $Pb_7Bi_4Se_{13}$, respectively. **b** Electronic DOS projected onto s and p orbitals for each element. Fermi surfaces of **c** electron-doped $Pb_7Bi_4Se_{13}$, and **d** hole-doped $Pb_7Bi_4Se_{13}$.

(DOS). In the region from the CBM to 1 eV above the Fermi level ($E_f$), the states are dominated by p orbitals of Pb, Bi, and Se, with minor components from s orbitals of Se. The valence bands ranging from 1 eV below the $E_f$ to the VBM is dominated by Se-4p and Pb-6s states with minor mixed contributions of Bi-6s, Bi-6p, and Pb-6p states. The $6s^2$ electrons of Pb and Bi are deeply located below Se 4p, forming lone-pair electrons. In the band structure shown in Fig. 2a, the orange and red dashed lines represent the $E_f$ calculated from DFT for the 0.05 electron-doped and 0.05 hole-doped $Pb_7Bi_4Se_{13}$. In the electron-doped $Pb_7Bi_4Se_{13}$, the $E_f$ goes across two bands at both $Y_2$ and $M_2$, which forms a nested Fermi surface consisting of two cylindrical-like electron sheets, visible in Fig. 2c. On the other hand, Fig. 2a shows that the Fermi level only lies in one band at both $Y_2$ and $M_2$ in the case of hole-doped $Pb_7Bi_4Se_{13}$, and the hole pocket at $M_2$ is obviously larger than $Y_2$, confirmed by the Fermi surface in Fig. 2d. It should be noted that nested bands could trigger strong electron–phonon scattering by the zone center phonons, which is a detrimental effect on charge transport. It is meaningful to calculate the scattering rates for each band and valley by using the electron–phonon coupling calculations based on density-functional perturbation theory, which will be the subject of our future work separately.

**Non-parabolic charge transport.** Parabolic band assumption is dominant for TE materials with wide bandgaps. For semiconductors with narrow band gaps, like PbTe[7] and $CoSb_3$[8], the non-parabolicity of electronic bands occurs, which stems from interactions between conduction and valence bands[8,39]. As shown in Fig. 3a, three-dimensional (3D) plots of parabolic and Kane bands are presented. The color bar is indicative of the relative energy levels. Clearly, the parabolic band reveals an ellipsoid-shaped energy surface. By contrast, the Kane band demonstrates a non-parabolic energy surface deviating from band edges. The conduction band nestification in parabolic and Kane bands are also considered. Two conduction bands are located at the same **k** point and flat valence bands are shown in the Brillouin zone. Figure 3b presents the two-dimensional (2D) dispersion relations, $E(\mathbf{k})$ ($E$ energy, **k** wave vector). With respect to the parabolic band, $E$ is proportional to $\mathbf{k}^2$, complying with the relationship, $E = \frac{\hbar^2\mathbf{k}^2}{2m^*}$. Here its effective mass, $m^*$ keeps as constant ($m^* = m_0^*$ effective mass at band edge). For the Kane band, it remains parabolicity at the energy extrema of electronic bands but deviates from parabolicity in a linear **k** tendency with departing from band edges. And the $E(\mathbf{k})$ dispersion could be well described by

$E(1 + \frac{E}{E_g}) = \frac{\hbar^2 k^2}{2m^*}$. Its effective mass, $m^*$ varies with Fermi level, following this relationship, $m^* = m_0^*(1 + \frac{2E}{E_g})$[8].

To capture the effective mass of $Pb_7Bi_4Se_{13}$ at room temperature, the pisarenko plot based on single Kane band (SKB) and single parabolic band (SPB) assumptions could be well-simulated by the effective mass, $m_s^*$ of 1.1 $m_e$, as suggested in Fig. 3c. The temperature dependence of effective mass verifies its band nonparabolicity, as shown in Fig. S10b. It should be noted that the Seebeck effective mass, $m_s^*$ is defined as the density of state effective mass in TE community, which predicts the Seebeck coefficient with Hall carrier concentration, $n_H$ in both the SPB and SKB models. However, it should be also realized that the qualitative difference between $m_s^*$ (Seebeck effective mass) and $m^*$ (effective mass from the $E \sim \mathbf{k}$ dispersion). The Seebeck effective mass is defined by Eqs. (1) and (2), which remain unchanged with increasing $n_H$. Differently, the effective mass, $m^*$ defined by $m^* = \hbar(d^2E/dk^2)^{-1}$, increases with increasing $E$. The temperature-dependent $n_H$ of two compositions of $(Pb_{0.98}Ga_{0.02})_7Bi_4Se_{13}$ (Ga1, $n_H = 2.0 \times 10^{20}$ cm$^{-3}$) and $(Pb_{0.95}Ga_{0.05})_7$-$Bi_4Se_{13}$ (Ga2, $n_H = 1.2 \times 10^{20}$ cm$^{-3}$) is presented in the inset of Fig. 3c. The $n_H$ of the two compositions remains unchanged up to 600 K. Meanwhile, the Hall mobility, $\mu_H$ of all compositions as a function of $n_H$ follows the Kane-type charge transport characteristic, demonstrated by the red solid line.

This $n_H$ dependent $\mu_H$ considering SKB model is calculated based on the non-degenerate mobility, $\mu_0$ at 300 K. This relationship is presented as below[40]

$$n_H = \frac{(2m_d^* k_B T)^{3/2}}{3\pi^2 \hbar^3} \cdot \frac{[(2K+1) \cdot {}^0F_{-2}^1]^2}{3K(K+2) \cdot {}^0F_{-4}^{1/2}}, \quad (1)$$

$$\mu_H = \frac{e}{\sqrt{2m_I^*}} \cdot \frac{\pi\hbar^4 v_l^2 d}{\Xi^2(m_b^* k_B T)^{3/2}} \cdot \frac{3K(K+2) \cdot 3^0F_{-4}^{1/2}}{(2K+1)^2 \cdot {}^0F_{-2}^1}, \quad (2)$$

where $k_B$ is the Boltzmann constant, $\hbar$ is the reduced Planck constant, $K$ is the anisotropy of Fermi surface, defined by $K = m_\parallel^*/m_\perp^*$, $m_\parallel^*$ and $m_\perp^*$ exhibit the longitudinal and transverse effective mass. $K$ is assumed to be 1 here. ${}^nF_k^m$ is the generalized Fermi integral. $v_l$ is the speed of sound, $d$ is the sample density, $m_I^*$ and $m_b^*$ are inert and single-valley effective mass.

**Unique quality factor analysis.** Conventional quality factor ($B$) analysis provides an insight into TE physics. It decouples $zT$ into the tunable reduced Fermi level, $\eta$ and $B$, based on the SPB model

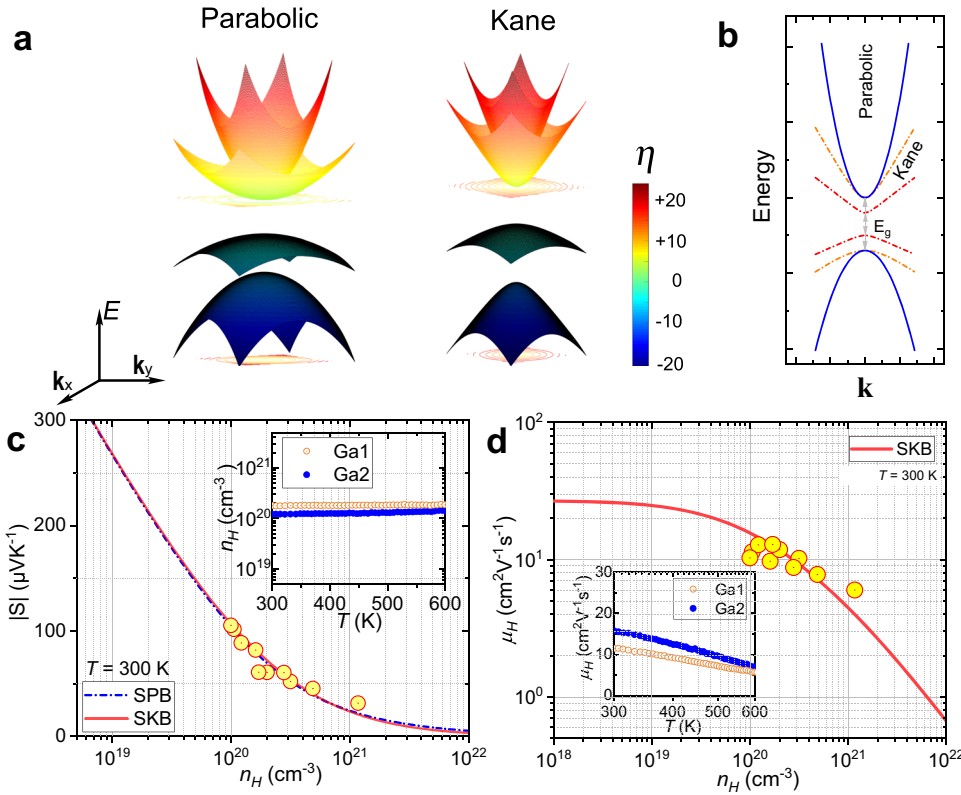

**Fig. 3 Electronic transport properties. a** 3D illustration of Parabolic and Kane bands with asymmetric conduction and valence bands. The features of conduction band nestification and relative flat valence band are presented. **b** $E \sim \mathbf{k}$ dispersions of parabolic (solid blue line) and Kane bands with different band gaps, $E_g$ (orange and red dot-dash lines). **c** Hall electron concentration, $n_H$ dependence of the magnitude of Seebeck coefficient (Pisarenko plot). The inset shows the temperature dependence of $n_H$ of two compositions, Ga1 (red circle) and Ga2 (blue circle). **d** Hall mobility, $\mu_H$ as a function of $n_H$. The red solid line is calculated based on the single Kane band (SKB) model. The inset shows the temperature dependence of $\mu_H$ of Ga1 and Ga2.

(see Supplementary Note 2). However, this SPB model is limited to fully describe TE semiconductors with band nonparabolicity and bipolar effect[41]. At first, the discernible difference between parabolic and Kane bands, shown in Fig. 3a, b, enables non-negligible deviations by using SPB to describe the electronic transport of non-parabolic bands. Secondly, the bipolar effect frequently emerges in semiconductors with small band gaps. This effect leads to a decrease in the Seebeck coefficient and increases in thermal conductivity, which severely deteriorates TE performances at high temperatures[41]. Obviously, this bipolar effect limits the applications of conventional quality factors. Thirdly, even taking advantage of the first-principles calculation, it is still time-consuming and expensive to determinate fundamental parameters, such as effective mass, carrier relaxation time, and deformational potentials of electronic bands. Consequently, an unconventional quality factor considering both band non-parabolicity and bipolar effect is highly necessary. Herein, this unique quality factor, $B^*_{\mathrm{Kane}}$ is developed by using two Kane band (TKB) models. Distinct from the traditional quality factor, the $zT$ here is closely correlated with the reduced Fermi level, $\eta$ ($\eta = (E - E_f)/k_B T$), and the reduced bandgap, $\xi$ ($\xi = E_g/k_B T$), which is summarized as follows

$$zT = \frac{(S_e \gamma + S_h)^2}{(\gamma + 1)\left[\frac{(k_B/e)^2 \gamma \xi}{3 B^*_{\mathrm{Kane}} \cdot {}^0 F^1_{-2}} + (L_e \gamma + L_h)\right] + \gamma(S_e - S_h)^2}, \quad (3)$$

And this unique quality factor $B^*_{\mathrm{Kane}}$ is generalized as below

$$B^*_{\mathrm{Kane}} = \frac{k_B}{e^2} \cdot \frac{\sigma_{E_0} E_g}{\kappa_L} \quad (4)$$

in which the $S$ denotes the Seebeck coefficient, $L$ is the Lorenz

number, $\sigma_{E_0}$ is the transport coefficient, $\gamma$ is the electrical conductivity ratio. The subscripts, $e$ and $h$, denote electrons and holes. The derivation is presented in Supplementary Note 3.

With the determination of physical parameters of the conduction band, we could obtain a series of $zT$ values by changing $B^*_{\mathrm{Kane}}$[42]. This $B^*_{\mathrm{Kane}}$ considers band nonparabolicity, band degeneracy $N_v$, inertial effective mass $m_I^*$, and deformation potential $\Xi$. An example is provided based on $Pb_7Bi_4Se_{13}$ with a specific $B^*_{\mathrm{Kane}} = 6$. Figure 4a depicts the theoretical $zT$ as functions of $\eta$ and $\xi$. A maximum $zT$ could be achieved with simultaneously optimized $\xi$ and $\eta$. A contour plot is presented in Fig. 4b with the $B^*_{\mathrm{Kane}}$ of 6. The progressive changes in the color bar from blue to red correspond to $zT$ from 0 to 1.48. Contour lines with representative $zT$ values, such as 0.4, 0.8, and 1.4, are marked by dash-dot lines in Fig. 4b. Taking $Pb_7Bi_4Se_{13}$ with $n_H$ of $1.2 \times 10^{20}\,\mathrm{cm}^{-3}$ as an example, its $\eta$ and $\xi$ at 800 K could be estimated to be $-0.5$ and 4.6, which is indicated by the yellow dot in Fig. 4b. The estimation of $\eta$, $\xi$, and $B^*_{\mathrm{Kane}}$ is presented in Supplementary Note 4. It is worth noting that the difference in conduction and valence band is considered and included in this model by using the band anisotropy, $\Lambda$.

A maximum $zT$ of 1.48 is predicted as represented by the red point in Fig. 4b. It can be achieved by further optimizing $\eta$ and $\xi$, indicated by the yellow arrow. For the tuning of bandgap, the chemical substitution of selenium by sulfur or tellurium tends to enlarge and decrease band gaps, respectively. This quality factor could be further optimized via chemical modifications on electronic band structures. The introduction of coherent nanoscale defects is favorable for improving quality factors, which could guarantee unchanged carrier mobilities, concomitantly strengthen

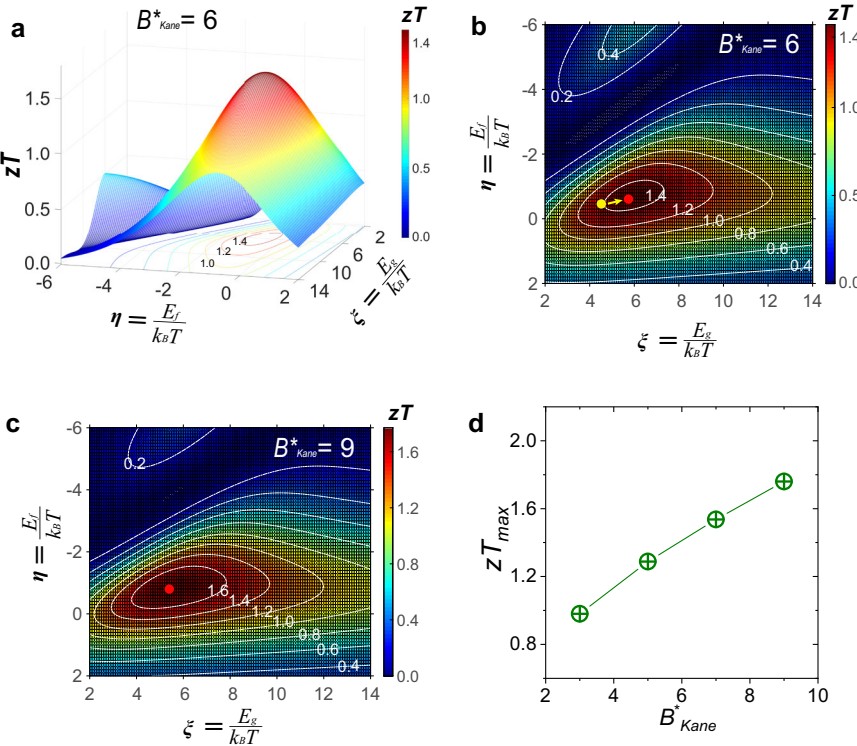

**Fig. 4 Quality factor analysis. a** 3D plot of figure of merit, $zT$ as functions of reduced Fermi level, $\eta$ and reduced band gap, $\xi$. **b** Contour plot of $\eta$- and $\xi$-dependent $zT$ with a quality factor, $B^{*}_{\text{Kane}} = 6$. The yellow dot corresponds to the experimental data of Pb$_7$Bi$_4$Se$_{13}$ with $n_H$ of $1.2 \times 10^{20}$ cm$^{-3}$ (Ga2). The red point indicates the highest $zT$ value. Contour lines in white color correspond to a series of $zT$ values. **c** Contour plot of $\eta$- and $\xi$-dependent $zT$ with $B^{*}_{\text{Kane}} = 9$. **d** The maximum $zT$ ($zT_{\text{max}}$) as a function of $B^{*}_{\text{Kane}}$.

defect-phonon scattering, and further decrease lattice thermal conductivity. Specifically, this could be chemically realized by introducing nanoscale precipitates such as SrSe into Pb$_7$Bi$_4$Se$_{13}$ host matrix, which is proven effective in the previous work[26]. There are still a variety of effective chemical methods to increase quality factors, such as decreasing band offsets to increase effective band convergence by doping heterogeneous elements at Pb or Bi sites. If $B^{*}_{\text{Kane}}$ turns into 9, the maximum $zT$ is predicted to be 1.76 as shown by the red point of Fig. 4c. Figure 4d shows that the maximum $zT$ is proportional to the enhanced $B^{*}_{\text{Kane}}$, which can be used as an insightful guide to exploiting high-performance TE materials.

It should be noted that the quality factor with SKB model has already been established, which could evaluate and predict the TE performance[40]. However, its application is limited at low temperatures, due to thermally excited minority carriers with increasing temperatures. The contribution of minority carriers becomes inevitable and considerably deteriorates TE performances. A lack of effective quality factors brings a great challenge to evaluate and predict TE performances in semiconductors with narrow band gaps, especially at high temperatures. To resolve this issue, this unique quality factor is established with TKB model, which can provide an effective and time-saving method. The derivations on previously reported quality factors established on SPB and SKB models ($B_{\text{Para}}$ and $B_{\text{Kane}}$ for clarity)[6,40], and developed $B^{*}_{\text{Kane}}$ are presented in Supplementary Note 5. With regard to previously reported quality factors, $zT$ values only correlate with the $B_{\text{Para}}$ (or $B_{\text{Kane}}$) and reduced chemical potential, $\eta$. By contrast, the $zT$ in this unique quality factor depends on three independent variables, $B^{*}_{\text{Kane}}$, $\eta$ and $\xi$. The 3D and contour plots of three different quality factors are presented in Fig. S12. This elaborately developed $B^{*}_{\text{Kane}}$ not only plays a significant role in evaluating and predicting TE performance in Pb$_7$Bi$_4$Se$_{13}$. It can

also be extended to more TE materials with narrow bandgaps, such as prototypical (Bi,Sb)$_2$Te$_3$[36] and van der Waals crystal Ta$_4$SiTe$_4$[43]. By adopting this $B^{*}_{\text{Kane}}$, the highest $zT$ of 1.54 and 0.27 are predicted in (Bi,Sb)$_2$Te$_3$ and Ta$_4$SiTe$_4$, which could be achieved by further optimizing the $\eta$ and $\xi$.

**Understanding low lattice thermal conductivity.** Lattice thermal conductivities, $\kappa_L$ of Pb$_7$Bi$_4$Se$_{13}$ samples are presented in Fig. 5a. The $\kappa_L$ is estimated by subtracting electronic thermal conductivity, $\kappa_e$ from total thermal conductivity. The $\kappa_e$ is estimated by using the Wiedemann–Franz equation, $\kappa_e = L\sigma T$. The $\kappa_L$ ranges from 0.44 to 0.29 Wm$^{-1}$ K$^{-1}$ at 300 K, and decreases with rising temperature. This decline deviates from the $T^{-1}$ tendency dominated by Umklapp phonon-phonon scattering, which implies strong defect-phonon scattering. Specifically, the $\kappa_L$ for Pb$_7$Bi$_4$Se$_{13}$ with $n_H$ of $1.2 \times 10^{20}$ cm$^{-3}$ decreases from 0.33 Wm$^{-1}$ K$^{-1}$ at 300 K to 0.17 Wm$^{-1}$ K$^{-1}$ at 800 K. This low $\kappa_L$ is comparable to that of state-of-the-art TE materials, such as 0.18 Wm$^{-1}$ K$^{-1}$ for CsAg$_5$Te$_3$[44], 0.13 Wm$^{-1}$ K$^{-1}$ for both (Ge,Mn, Sb)Te[45] and Bi$_{1-x}$Pb$_x$CuSeO[46]. To further assess the intrinsic Umklapp phonon interaction, sound speeds were measured. The related physical parameters were calculated and tabulated in Table S6. The mean sound speed for Pb$_7$Bi$_4$Se$_{13}$ is as low as 1553 ms$^{-1}$, and the Grüneisen parameter $\gamma_G$ is 2.2. This large $\gamma_G$ is indicative of soft chemical bonding and strong lattice vibration anharmonicity, which is comparable with $\gamma_G$ of 2.6 for Ag$_9$GaSe$_6$[19], 2.1 for AgSbTe$_2$[47], 1.7 for K$_2$Bi$_8$Se$_{13}$[48], 1.7 for BiSe[49], and 1.6 for Cu$_{17.6}$Fe$_{17.6}$Se$_{32}$[50], etc. The calculated Debye temperature ($\Theta_D$) is as low as 148 K, comparable to 147 K in Ag$_9$GaSe$_6$ with liquid-like thermal conductivity[33]. The large Grüneisen parameter and low Debye temperature lead to large lattice anharmonicity and strong Umklapp phonon–phonon scattering.

The low-temperature heat capacity in Fig. S13 shows the non-linear relationship of $C_p/T$ vs. $T^2$. This inconsistency with the conventional Debye model implies the existence of localized

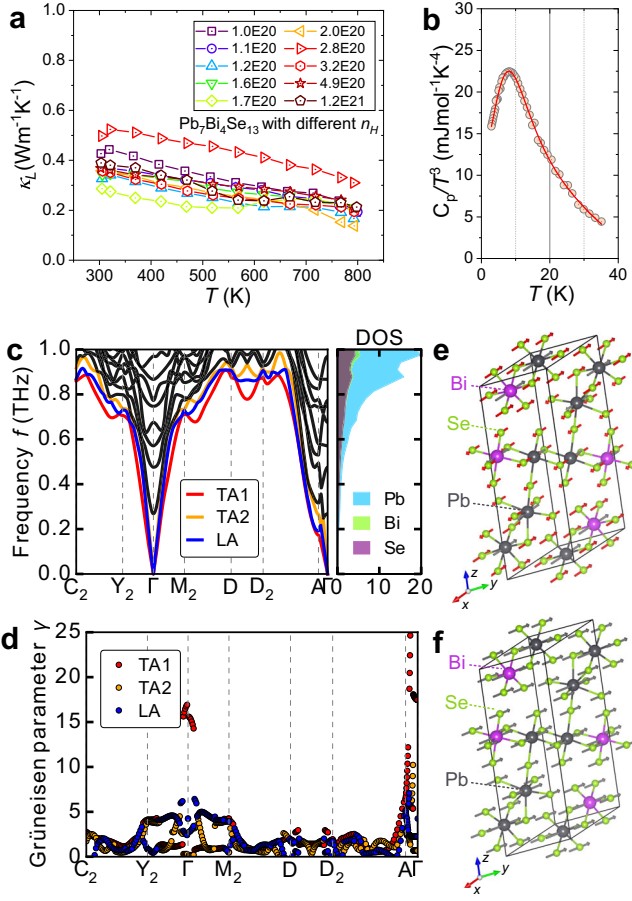

**Fig. 5 Lattice thermal conductivity and phonon-related calculations.**
**a** Lattice thermal conductivity, $\kappa_L$. **b** $T$ dependence of $C_p/T^3$. The circle is the experimental data and the line is established by using a Debye and three Einstein modes. **c** Phonon dispersions and atom-projected phonon DOS of the low-frequency phonons up to $f = 1.0$ THz. Transverse acoustic phonons, TA1 and TA2 are plotted in red and orange colors. Longitudinal acoustic phonon, LA, is in the blue line. Optical phonons are in black lines. In the low-frequency region, the dominant contribution stems from Pb atoms. **d** Grüneisen dispersions. Displacement patterns of **e** the low-frequency optical phonon at Γ point, and **f** low-frequency TA1 mode at Γ point.

Einstein oscillator modes. A broad maximum appears around 8 K in the $T$ dependence of $C_p/T^3$ (Fig. 5b). This additional contribution to heat capacity is a signature of localized optical phonon modes, frequently found in TEs with low thermal conductivities[17,51]. To quantitatively describe the heat capacity, a combined Debye–Einstein model is adopted (see Table S7). Three low-frequency Einstein vibration modes are utilized for a full description of the measured data. And the Einstein characteristic temperatures are $\Theta_{E1} = 13$ K, $\Theta_{E2} = 33$ K, and $\Theta_{E3} = 119$ K, respectively.

To understand the origin of low thermal conductivity, first-principles phonon calculations were performed to study lattice dynamics. Phonon dispersions and atom projected phonon DOS in the low-frequency region are presented in Fig. 5c. The full phonon spectrum and DOS is demonstrated in Fig. S14. Intriguingly, optical phonon modes with low energies at Γ point can be observed around 0.28, 0.45, and 0.59 THz, respectively. These lowest optical phonons are in good agreement with experimentally verified lowest Einstein modes observed in Fig. 5b. Actually, the low-energy vibrational modes in phonon DOS have been directly observed in $(PbSe)_5(Bi_2Se_3)_{3m}$ by the inelastic

neutron scattering measurement[52], which shares similar structural complexity and compositions with $Pb_7Bi_4Se_{13}$. Besides, the maxima centered below 10 K in the relationship of $T \sim C_p/T^3$ have also been observed in $(PbSe)_5(Bi_2Se_3)_{3m}$. These features have also been reported in compounds with low thermal conductivities, such as $BaGa_5$ and $InTe$[53,54]. The low-lying optical phonons couple with heat-carrying acoustic phonons, decreasing group velocities and thereby restraining heat transport. Similar low-frequency optical phonons have been observed in typical TE materials with usually low $\kappa_L$, such as $MgAgSb$[20], $TlInTe_2$[51], and $AgBi_3S_5$[18]. Atom-projected phonon DOS shows the dominant contribution from Pb atoms in the low-frequency region (<1.0 THz), strongly correlated to its $6s^2$ lone pair electrons[31,55]. The specific vibration pattern of this lowest optical phonon at Γ point is visualized in Fig. 5e.

According to $\kappa_L \sim \gamma_G^{-2}$, $\gamma_G$ is a measure of anharmonicity of phonon modes and is inversely proportional to lattice thermal conductivity. To quantitively evaluate lattice anharmonicity, we plot the Grüneisen dispersions of acoustic phonons in Fig. 5d. A striking feature is the anomaly large values of $\gamma_G$ on A-Γ, $Y_2$-Γ, and $M_2$-Γ paths, indicating largely enhanced anharmonicity. Specifically, the $\gamma_G$ of TA1 at Γ point reaches up to a high value of 16. Such high $\gamma_G$ indeed demonstrates strong anharmonicity and intensified the Umklapp phonon-phonon scattering. And this specific vibration pattern is depicted in Fig. 5f. We further used the quasi-harmonic approximation methods to calculate the mode-averaged $\gamma_G$ to be 2.0, in good agreement with the experimental $\gamma_G$ of 2.2. Therefore, the low thermal conductivity has an intimate relationship with localized low-frequency optical phonon modes and the large anharmonicity of acoustic phonons. The SOC effect on phonon calculation has also been considered, which demonstrates a limited effect on the calculated phonon spectra (see Fig. S14).

**Coexistence of diverse defects.** Nanoscale defects also play a significant role in the low thermal conductivity. To elucidate defect-phonon scattering sources, a microstructural investigation by using a spherical aberration-corrected scanning transmission electron microscope (Cs-corrected STEM) was employed on $Pb_7Bi_4Se_{13}$ with $n_H$ of $1.2 \times 10^{20}$ cm$^{-3}$. High-angle annular dark-field (HAADF)-STEM images in Fig. 6a, b clearly show the periodic defective stripes, with a length beyond 30 nm and a width of ~3 nm, embedded in the host matrix of $Pb_7Bi_4Se_{13}$. The defective stripes include high-density of stacking faults, shown by the white arrows in Fig. 6b. The elemental mapping in Fig. S15 verifies the homogeneously distributed elements along with defective stripes. The strain map profiles ($\varepsilon_{xx}$) in Fig. 6a1, b1 are derived by geometric phase analysis (GPA), which semi-quantitatively evaluate spatially distributed strain fields. Intriguingly, the periodic strain lines are captured (labeled by black arrows) and demonstrate the intense strains along with these defective stripes in Fig. 6a, b. The strain analysis of $\varepsilon_{yy}$ is presented in Fig. S15. In addition, dislocation cores are also observed, highlighted by white circles. Figure 6c presents the irregularly shaped nanoscale precipitates embedded in the host matrix, marked by the blue dotted circles. These high number density of nanoscale precipitates demonstrates the size of several nanometers. Diverse lattice imperfections have been evidently observed in Fig. 6d. The moiré fringes, marked by the yellow ellipses, distribute around the dark-contrast precipitates. These strong moiré fringes originate from the interference between different sets of lattice planes, indicating local mass and strain fluctuations. Figure 6e shows the enlarged region from the inverse fast Fourier transform of Fig. S16c. In sharp contrast to the slightly distorted lattice in Fig. 6e, another two regions in Fig. 6f, g

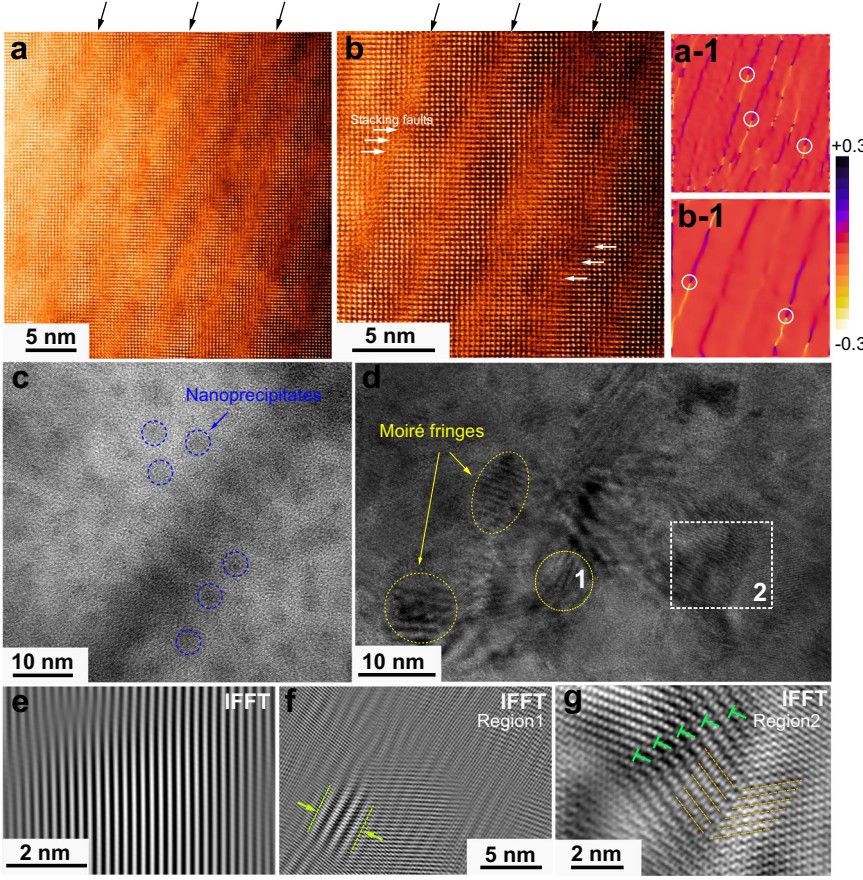

**Fig. 6 Nanoscale defects. a, b** HAADF-STEM images showing periodic defective stripes, marked by black arrows. High-density stacking faults are observed and indicated by white arrows. **a-1, b-1** Corresponding GPA results ($\varepsilon_{xx}$) of (**a**) and (**b**), revealing intense lattice strains. The periodic strain lines are indicated by black arrows, in which dislocation cores are marked by white circles. **c** HRTEM image showing the high number density of nanoscale precipitates, encircled by blue dotted lines. **d** TEM image demonstrating strong moiré fringes indicated by yellow ellipses and highly distorted lattice marked by a white square. Inverse fast Fourier transform (IFFT) images of selected regions. **e** Defect-rare region, **f** defect-rich region 1, full of moiré fringes, and **g** region 2, filled up with dislocation arrays and twinned grain boundaries.

highlight the crystal imperfections. In Fig. 6f, moiré patterns are highlighted by the pleated structure, marked by the yellow lines. In Fig. 6g, the coexistence of dislocation arrays and nanoscale twined structures, demonstrated by green symbols and parallel brown lines, respectively. Numerous nano-precipitates and dislocations can also be observed in Fig. S16. The coexistence of periodic defective stripes, nanoprecipitates, dislocation, and Moiré fringes play a significant role in scattering phonon with different length scales, which strongly suppresses lattice thermal conductivity.

## Discussion

In this work, we report the discovery of a unique TE material, $Pb_7Bi_4Se_{13}$ based lillianites. A high peak $zT$ of 1.35 at 800 K and a decent average $zT$ of 0.92 from 450 to 800 K are achieved for $n$-type $(Pb_{0.95}Ga_{0.05})Bi_4Se_{13}$. This is the highest TE performance realized in lillianites. For $Pb_7Bi_4Se_{13}$, conduction bands are highly nested at $Y_2$ and $M_2$ points. Besides, these nested conduction bands at $Y_2$ and $M_2$ also share similar energies, demonstrating a strong signature of band convergence. The synergistic effect of band nestification and convergence leads to higher band degeneracy and superior electrical properties. It provides an alternative approach to improve valley degeneracy for low-symmetry TE systems. These electronic band features are rarely documented in compounds with lower crystal symmetries and are comparable to typical $n$-type TE materials, such as PbTe and $CoSb_3$, which both

demonstrate multiple degenerated bands as well as narrow-band gaps ($E_g = {\sim}0.3$ eV for PbTe and $\sim$0.23 eV for $CoSb_3$)[7,8]. For PbTe, conduction bands consist of one Kane band at the L point. The full valley degeneracy, $N_v$ for L are 4. For $CoSb_3$, conduction bands are composed of one Kane band at $\Gamma$ point ($N_v = 3$) and one parabolic band along the $\Gamma$–N line ($N_v = 12$).

The conventional quality factor neglects the features of band nonparabolicity and bandgap, whose application is severely limited in narrow-gap TE materials. A unique quality factor, $B^*_{Kane}$ is deliberately established here, which can break previous limitations and provide an effective method to predict and evaluate TE properties. Furthermore, a low lattice thermal conductivity, $0.17\ Wm^{-1}\,K^{-1}$ is achieved at 800 K. The intrinsic and extrinsic phonon scattering sources synergistically suppress phonon transport and achieve the ultralow lattice thermal conductivity. The intrinsic component stems from strong anharmonicity of an acoustic phonon with large Grüneisen parameter ($\gamma_G = 16$), localized low-frequency optical phonons (<0.6 THz), and low Debye temperature ($\Theta_D = 148$ K). Moreover, the extrinsic nanoscale lattice imperfections, including periodic defective stripes, nano-precipitates, dislocations, and Moiré fringes, facilitate to impede heat transport. This work could rekindle the hope and vision of lillianites as promising high-performance TE materials. Furthermore, the unique quality factor paves an attractive route to accelerate the exploration for TE compounds with resembling band characteristics.

## Methods

**Synthesis**. Pb shots (99.99%), Ag shots (99.999%), Ga chunks (99.999%), In shots (99.999%), Se shots (99.999%), and I pieces (99.999%) were purchased from Sigma-Aldrich. Firstly, precursors of $Pb_{1-x}M_x$Se (M = Ga, In or Ag), $PbSe_{1-x}I_x$, and $Bi_2Se_3$ were prepared by loading the raw materials into vacuum quartz tubes according to the stoichiometric ratio, then heated up to 1373 K in a box furnace. Ingots of $(Pb_{1-x}Ga_x)_7Bi_4Se_{13}$, $(Pb_{1-x}In_x)_7Bi_4Se_{13}$, $(Pb_{1-x}Ag_x)_7Bi_4Se_{13}$ and $Pb_7Bi_4Se_{13-x}I_x$ were prepared by loading precursors $Pb_{1-x}M_x$Se (M = Ga, In or Ag) or $PbSe_{1-x}I_x$ and $Bi_2Se_3$ into ball-milling vials in a glove box ($O_2$ and $H_2O$ content < 0.1 ppm), according to the normal compositions. Precursors were mechanically alloyed by high-energy ball-milling (SPEX 8000D). To ensure full and homogeneous milling, the chemicals attached to the inner wall of vials were scratched out in the glove box after ball-milling for 1.5 h. This process was repeated eight times. After the long-term ball-milling, the powders were pressed into pellets in the glove box. The pellets were subsequently flame-sealed into vacuum quartz tubes. Then, the quartz tubes were transferred into a box furnace and heated to 573 K in 24 h, dwelled for 72 h, followed by heating up to 850 K in 24 h, and then soaked for two weeks. Subsequently, they were cooled down to room temperature in 12 h. These ingots were crushed into powders by mortar and pestle in a glove box and densified by an SPS system (Ed-PasIVJ, 6T-3P-30, Japan) at 823 K for 10 min with an axial pressure of 75 MPa under vacuum. The sintered pellets were then cut into a $3 \times 3 \times 12$ mm$^3$ rectangular bar and into a $\varnothing 12.7 \times 2$ mm disk for measurements.

**Characterization**. Laboratory X-ray patterns were collected on PANalytical X'pert Pro powder diffractometer with Cu target (operating under 40 kV and 30 mA). The electrical conductivity and Seebeck coefficient were measured simultaneously using ZEM-3 system (ULVAC-RIKO, Japan). The thermal conductivity was calculated by $\kappa = DC_p\rho$, in which the Dulong–Petit specific heat was taken as the specific heat ($C_p$), the pellet density ($\rho$) was calculated by the geometrical method and the thermal diffusivity ($D$) was measured using the lasher flash method (DLF 1200, TA Instruments, USA). Room temperature charge carrier concentration was determined from the Hall coefficient measurement using the Van der Pauw method (Bio-Rad Microscience, Hall measurement system HL5500, USA). High-temperature carrier concentration, Hall coefficient and Hall mobility were collected at the homemade device with the Van der Pauw method. Cross-sectional transmission electron microscopy (TEM) samples were prepared using the conventional in situ lift-out method, which was performed on a dual-beam scanning electron microscope-focused ion beam (SEM-FIB) instrument (Crossbeam XB540, Zeiss, Germany) with an attached micromanipulator (OmniProbe 400, Oxford Instrument, UK). High-angle annular dark-field scanning transmission electron microscopy (HAADF-STEM) images were taken in an aberration-corrected JEOL ARM-200F operating at 200 kV. The collection angle of the HAADF-STEM image was from 80 to 250 mrad. EDS mapping was carried out by an Oxford X-Max EDS detector. The microstructures were characterized by a field-emission gun transmission electron microscope (FEG-TEM, JEM-2100F, JEOL, Japan). TEM and HRTEM images were taken using a Gatan Ultrascan 1000XP CCD. The samples for the compressive test, with a size of $\varnothing 2 \times 4$ mm, were machined using an electron discharge machining machine, conducted by an Instron 3384 Electro-mechanical Universal Testing Machines. The nanoindentation investigation was performed on Hysitron TI-980 Triboindenter nanoindenter (Bruker Nano Surfaces, USA).

**Electronic structure and phonon calculations**. First-principles density functional theory (DFT) calculations were performed using the Vienna Ab-initio Simulation Package[56,57]. The generalized gradient approximation in the Perdew–Burke–Ernzerhof parametrization[58] was used as the exchange-correlation functional. The cutoff energy was set to 400 eV which has been sufficient to guarantee accuracy. We used a primitive cell of $Pb_7Bi_4Se_{13}$ including 24 atoms in modeling the electronic properties. The energy convergency threshold and force convergency threshold in the structural optimization were $10^{-8}$ eV and $10^{-4}$ eV Å$^{-1}$, respectively. A **k**-mesh of $9 \times 9 \times 3$ was used to sample the **k** points in the Brillouin zone. The rigid band approximation was used to model the n-type and p-type $Pb_7Bi_4Se_{13}$. Specifically, the electron/hole doping in $Pb_7Bi_4Se_{13}$ is achieved by adding extra electrons/holes to the system with the same amount of uniform positive/negative charge in the background. Within the framework of rigid band approximation, we find the overall shape of band structures are almost unchanged except the Fermi level shift, hence we only show the calculated Fermi level of 0.05 electron- and hole-doped $Pb_7Bi_4Se_{13}$ in Fig. 2a of the main text. The corresponding Fermi surfaces were calculated by using a very dense $45 \times 45 \times 15$ k-mesh to ensure the accuracy of calculations for metallic states. To evaluate the lattice dynamic properties, we used the finite displacement supercell method to calculate the phonon spectrum and phonon density of states and used the quasi-harmonic method to calculate the Grüneisen parameter. Both the finite displacement supercell method and quasi-harmonic method implemented in Phonopy[59] were used. In addition, the finite displacement supercell method was used with the $3 \times 3 \times 1$ supercell (216 atoms) of the primitive cell with a finite displacement of 0.1 Å. The crystal structures and displacement patterns were all visualized using VESTA software[60]. When computing the phonon density of states of mode-averaged Grüneisen, we used a q-mesh of $20 \times 20 \times 20$.

## Data availability

The authors declare that all data supporting the findings of this work are available from the corresponding authors upon reasonable request.

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

## Acknowledgements

Q.Y.Y. acknowledges the Singapore MOE Tier 2 under Grant MOE2018-T2-1-010, Singapore A*STAR Pharos Program SERC 1527200022. Q.Y.Y. and J.W.X. acknowledge the Singapore A*STAR project A19D9a0096. This work is also supported by the Japan Society for the Promotion of Science (JSPS) KAKENHI, Grant JP 19F19057. L.H. acknowledges the International Research Fellowship of JSPS. F.Y.Q. acknowledges the Chinese Scholarship Council (CSC) for the scholarship in Tokyo Institute of Technology. X.X.Z. thanks for the support from the Presidential Postdoctoral Fellowship, Nanyang Technological University, Singapore via grant 03INS000973C150. K.H. and D.V.M.R. acknowledge funding from the Accelerated Materials Development for Manufacturing Program at A*STAR via the AME Programmatic Fund by the Agency for Science, Technology and Research under Grant No. A1898b0043. The electron microscopy and XRD work were performed at the Facility for Analysis, Characterization, Testing, and Simulation (FACTS), Nanyang Technological University, Singapore. Y.-W. Fang acknowledges the computational resources provided by the New York University New York, Abu Dhabi, and Shanghai. L.H. acknowledges Prof. Sergey Nikolaev for the discussion and efforts on theoretical calculations.

## Author contributions

L.H. and Q.Y.Y. conceived this work. L.H., Y.B.L., and A.S. prepared the samples and performed thermoelectric measurements. L.H. developed the unique quality factor. Y.-W.F. completed DFT calculations and interpreted the theoretical data. F.Y.Q. analyzed charge transport data by using physical models. X.C., X.X.Z., Y.Z.H., and Z.L. performed TEM characterization. D.V.M.R. and K.H. collected the low-temperature thermal conductivity, heat capacity, and Hall data. W.B.L. performed the mechanical test and low-temperature Hall measurement. U.A. helped for measuring sound speeds. T.S. and G.J.S. helped to collect high-temperature Hall data. L.H., J.W.X., and Q.Y.Y. wrote this paper. All authors reviewed, discussed, and approved the results and conclusions.

## Competing interests

The authors declare no competing interests.
