## [Peer Review File · Nature Communications]

REVIEWER COMMENTS

Reviewer #1 (Remarks to the Author):

The authors present the study of thermoelectric properties in $\text{Pb}_7\text{Bi}_4\text{Se}_{13}$ based lillianites that has been overlooked in the past. The authors propose a quality factor to account for the band nonparabolicity and bipolar effect. The topic is timely and important, not just to understand the materials under consideration but in general to understand the optimization of thermoelectric figure-of-merit in lillianite materials. I have three comments and one question:

- 1) Within the Kane model, the valence band is a mirror image of the conduction band. However, the DFT calculation clearly shows very different dispersions of the two bands. The proposed quality factor is based on the Kane model, thus leading to an underestimated bipolar effect and overestimated ZT.
- 2) The nested bands also introduce stronger electron-phonon scattering by the zone center phonons, a detrimental effect for the electrical conductivity. If the authors can calculate scattering rates for each band and valley, this would make the paper more solid.
- 3) Pb and Bi are massy elements and have strong spin-orbit coupling, which affects the band structure and the phonon dispersions. I don't see the authors mention that the spin-orbit coupling is included in the calculations. This should be clarified.
- 4) What are concrete ways in experiment to tune the bandgap and increase the quality factor in the considered materials?

Reviewer #2 (Remarks to the Author):

In this manuscript, the authors report on a joint experimental and theoretical investigation of the thermoelectric properties of the layered compound $\text{Pb}_7\text{Bi}_4\text{Se}_{13}$ doped with various elements on the Pb sites. A central aspect of the manuscript is the discovery of high dimensionless thermoelectric figure of merit ZT ascribed to the band convergence of several conduction bands driven by the dopants introduced. The experimental and theoretical results are sound and certainly deserve to be published somewhere. However, I do not find that this manuscript provides a sufficient level of novel insights or thought-provoking ideas to be publishable in Nature Communications. As mentioned by the authors in their introduction, related compounds in various similar homologous series have been studied for their thermoelectric properties. In addition, the beneficial role played by the convergence of several electronic bands on the thermoelectric properties have been widely discussed already in the literature. Adding another example of such behaviour is certainly interesting but appears insufficiently novel. I think the present manuscript would be a better fit for Communication Materials or Scientific Report, provided that the following remarks/issues have been carefully addressed by the authors.

- 1). This material belongs to the larger groups of homologous series based on Pb, Bi and Se, which have been already (at least for some of them) investigated for their interesting thermoelectric properties. In my opinion, the list of references given by the authors lack numerous relevant papers on these compounds. In addition, the authors may put into a broader perspective these materials by mentioning that other series, structurally related to lillianites, also exist (see for instance the following reference and the references therein: Lu et al. Mater. Adv. DOI: 10.1039/d0ma00912a (2021)).
- 2). One major technical issue in this manuscript is the lack of measurements performed parallel and perpendicular to the pressing direction. This compound crystallizes with a monoclinic crystal structure (a drawing of the crystal structure should be shown in the manuscript) and, as widely shown in the literature, polycrystalline samples exhibit significant anisotropy between these two directions. Thus, the lack of such measurement and discussion is unacceptable and not serious. Unfortunately, I believe

that the high ZT values reported in this manuscript are enhanced due to an erroneous combination of measurements in two different directions for the electrical and thermal properties. The authors should therefore show measurements performed in both directions (both Hall effect and transport properties) to strengthen their claim of high ZT values achieved.

3). In Figure 3, the authors show the Pisarenko plot together with the theoretical line calculated using a single-non-parabolic band model. In principle, for non-parabolic bands, the density of states effective mass (m^*) should vary with the carrier concentration. In the present case, even spanning one order of magnitude does not result in significant variations in m^* . May the authors explain in detail the reasons? The authors may show a theoretical line calculated with a single-parabolic band model to strengthen their claim. In addition, the values of K in equations (1) and (2) used for these calculations should be mentioned as well as the way they have been determined. Do the results presented depend on the K values? The same questions apply for the inertial mass. As a minor point, the $(n,m,k)F$ functions are not the Fermi integrals (that appear in the SPB model) but rather the generalized Fermi integrals.

4). The quality factor calculated for non-parabolic bands has already been done for PbTe-based compounds. The authors should discuss similarities/differences between these well-known thermoelectric compounds and the present lillianite sample.

5). Several typos are present in the manuscript and should be corrected, for instance, "Lorentz" should be "Lorenz", etc....

6). Regarding the lattice dynamics of this compound, a rapid survey of the literature indicates that an experimental study of the lattice dynamics of the $(\text{PbSe})_5(\text{Bi}_2\text{Se}_3)_3$ compounds has been published (Sassi et al., Phys. Chem. Chem. Phys. 20, 14597 (2018)). The authors should mention at minima this reference and discuss/compare their results with those obtained on these related materials.

Reviewer #3 (Remarks to the Author):

In the introduction, the authors present their hypothesis for bringing up the thermoelectric performance of doped $\text{Pb}_7\text{Bi}_4\text{Se}_{13}$ to an exceptional level. They list favorable parameters that the material exhibits, such as bond anharmonicity, which they later quantify through phonon calculations, and the existence of lone electronic pairs, all reducing the thermal conductivity.

To overcome the limitations of poor electrical conductivity in $\text{Pb}_7\text{Bi}_4\text{Se}_{13}$, they dope the material with Ga, In, Ag, and I, searching for the best dopant candidate. Out of the dopants, Ga in the composition $(\text{Pb}_{0.95}\text{Ga}_{0.05})_7\text{Bi}_4\text{Se}_{13}$ produces the best thermoelectric figure of merit of 1.35 at 800 K. The peak figure of merit tapers off very gently for lower temperatures. Through density functional calculations, they demonstrate that $\text{Pb}_7\text{Bi}_4\text{Si}_{13}$ has a flat valence band top and valley-degenerate conduction band bottom, both sought-after features in thermoelectric materials.

Moreover, the authors propose a new quality factor that considers the band non-parabolicity and the bipolar effect, as applicable to $\text{Pb}_7\text{Bi}_4\text{Se}_{13}$, in analyzing the figure of merit. This new quality factor predicts even potentially higher ZT for $\text{Pb}_7\text{Bi}_4\text{Se}_{13}$ that might be realized in the future.

In the last part of the paper, the authors, based on the TEM image analysis, identify various dislocations responsible for further reduction of the lattice conductivity.

This paper falls in the top tier of the thermoelectric research in the term of novelty and completeness. The results outperform the earlier reports on similar material (Ref. 43). Many similar research papers either lack complete mechanistic characterizations and discussions or do not present materials with a stellar performance. As a result, I think this paper can be published with a little more enhancement to the arguments presented. My recommended revisions are listed below:

1. What application do the authors think this material is best for? Industrial waste energy recovery at large scales? Or more minor applications such as in automobile parts?

2. Although $\text{Pb}_7\text{Bi}_4\text{Se}_{13}$ is thermoelectrically fully characterized, nothing was mentioned about the materials' thermomechanical properties. For instance, how robust are $\text{Pb}_7\text{Bi}_4\text{Se}_{13}$ and its doped variants towards successive cooling and heating cycles in terms of crack formation and detrimental ionic diffusion?

(A more thorough literature search may provide some answers to the questions above that can be included in the discussion part.)

3. Furthermore, as I collect, the concept of the "unique quality factor" is developed in this paper for addressing both band non-parabolicity and bipolar effect, which are very difficult for the earlier definition of quality factor (B Factor) to evaluate. Since this is a new concept, can the authors demonstrate its applicability for different materials in the SI, or at least prove that it converges to B Factor if solved for a parabolic band?

4. Finally, the manuscript needs a bit of polishing. For instance, "optimization on" on line 62 should be "optimization of." The punctuation after Equations 3 and 4 can be improved.

Response to the reviewer #1:

The authors present the study of thermoelectric properties in $\text{Pb}_7\text{Bi}_4\text{Se}_{13}$ based lillianites that has been overlooked in the past. The authors propose a quality factor to account for the band nonparabolicity and bipolar effect. The topic is timely and important, not just to understand the materials under consideration but in general to understand the optimization of thermoelectric figure-of-merit in lillianite materials. I have three comments and one question:

Comment 1: Within the Kane model, the valence band is a mirror image of the conduction band. However, the DFT calculation clearly shows very different dispersions of the two bands. The proposed quality factor is based on the Kane model, thus leading to an underestimated bipolar effect and overestimated ZT.

Response: Thanks for this comment. According to the DFT calculation, the dispersion of conduction band (CB) and valence band (VB) is obviously different. The previous Figure 3a only aims to highlight the difference of parabolic and Kane bands with symmetric CB and VB. However, it seems to easily result in misleading version that the symmetric CB and VB is also adopted in the unique quality factor model. Here we update this Figure 3a by using the asymmetric CB and VB. The conduction band nestification and flat VB are considered in this revised Figure 3a-3b.

With the regard to the two Kane band (TKB) model shown in Figure 4 of the main text, we indeed consider the difference of conduction and valence bands. The parameter λ is the electronic band asymmetry, which indicates the difference of conduction and valence bands in light of effective mass (m^*), band degeneracy (N_v) and deformation potential (\mathcal{E}). This value is estimated according to the different weighted mobilities at 800 K, and adopted in the two Kane band (TKB) model.

As regard to the weak bipolar effect, it is probably attributed to the increased band gap due to lattice expansion with increasing temperature, which suppress the contribution of minor carrier in this compound. For clarity, we also add this corresponding discussion in Figure 3a and Figure 4, highlighted in yellow.

Figure 3 Electronic transport properties. (a) 3D illustration of Parabolic and Kane bands with asymmetric conduction and valence bands. The features of conduction band nestification and relative flat valence band are presented. (b) $E \sim k$ dispersions of parabolic (solid blue line) and Kane bands with different band gaps, E_g (orange and red dot-dash lines).

The following statement is added into the main text (Page 15).

The conduction band nestification in parabolic and Kane bands are also considered, in which two conduction is located at the same k point of the Brillouin zone. In addition, the flat valence band is also presented, as shown in Figure 3a-3b.

Comment 2: The nested bands also introduce stronger electron-phonon scattering by the zone center phonons, a detrimental effect for the electrical conductivity. If the authors can calculate scattering rates for each band and valley, this would make the paper more solid.

Response: Thanks for this suggestion. We agree with this point that the band-resolved or valley-resolved scattering rates can help understand the effect of enhanced electron-phonon scattering in nested-bands on electrical conductivity. To get the electron-phonon scattering rates (the inverse of the relaxation time $\tau_{n\vec{k}}$), we have to obtain the electron self-energy because the relaxation time $\tau_{n\vec{k}}$ is connected to the imaginary part of the electron self-energy by $\frac{1}{\tau_{n\vec{k}}(\omega, T)} = 2\text{Im}\Sigma''_{n\vec{k}}(\omega, T)$ where ω and T are the phonon frequency and temperature. However, the electron self-energy, directly associated to the electron-phonon matrix element, is unavailable in our standard electronic structure calculations within the framework of density functional theory (DFT). It may not be realized in the lattice dynamics phonon calculations within the framework of supercell and finite displacement methods because the electron-phonon matrix element has to be solved from the very time-consuming electron-phonon

coupling calculations (e.g., based on density-functional perturbation theory). Consequently, this investigation on scattering rates on each band or valley is meaningful yet highly challenging for the current computational methods used in this study. A theoretical study focusing on the scattering rates of each band and valley using the electron-phonon coupling calculations will be the subject of our future work separately.

This following discussion is added into the main text (Page 13).

It should be noted that band nestification could trigger strong electron-phonon scattering by the zone center phonons, which is detrimental effect for charge transport. It is meaningful to calculate the scattering rates for each band and valley by using the electron-phonon coupling calculations based on density-functional perturbation theory, which will be the subject of our future work separately.

Comment 3: Pb and Bi are massy elements and have strong spin-orbit coupling, which affects the band structure and the phonon dispersions. I don't see the authors mention that the spin-orbit coupling is included in the calculations. This should be clarified.

Response: We thank reviewer #1 for bringing us attention to the spin-orbital coupling (SOC) effect on the band structure and the phonon dispersions. To address this issue, we appended the computationally heavier SOC-included electronic and phonon calculations.

Figure S9. Electronic band structure of $\text{Pb}_7\text{Bi}_4\text{Se}_{13}$ calculated by (a) standard DFT without SOC and (b) DFT with SOC. Fermi level is set to zero.

The following statement is appended into the main text (Page 12).

The spin-orbital coupling (SOC) effect on the band structure is also considered. The introduction of SOC has an indiscernible effect on the band dispersions of CBM

and VBM, as shown in Figure S9. The calculated E_g is suppressed from 0.67 eV (without SOC) to 0.22 eV (with SOC).

The following statement and Figure S9 are appended into the supplementary information (Page S13).

Figure S9 shows the band structures calculated by DFT without SOC (*i.e.*, Fig. 2a in the main text) and with SOC. Although the SOC is found to suppress the band gap from 0.67 eV (without SOC) to 0.22 eV (with SOC), it has discernable effect on the band dispersions near the conduction band minima (CBM) and valence band maxima (VBM) as highlighted by the orange squares, indicating that the low-energy electronic states do not be affected significantly with the inclusion of SOC.

The following statement is also added into the supporting information (Figure S14, Page S32).

Figure S14. Calculated phonon dispersions and phonon density of states (DOS) of $\text{Pb}_7\text{Bi}_4\text{Se}_{13}$ (a) without SOC and (b) with SOC.

To investigate the SOC effect on phonon dispersion, we also performed the phonon calculation without and with SOC. Figure S14 shows the phonon dispersions and phonon density of states (DOS) calculated from the finite-displacement and

supercell method (48 supercells in each calculation). We can find that the SOC effect has small effect on the overall dispersions. The Γ -point phonons in both calculations show the largest frequency of around 5 THz. In particular, the SOC effect on the low-frequency phonon dispersions are insignificant. Therefore, the effect of SOC on phonon transport of $\text{Pb}_7\text{Bi}_4\text{Se}_{13}$ should be limited, considering that the low thermal conductivity is mainly dominated by the low-frequency phonons.

Combining the appended electronic band calculations including SOC with the phonon calculations including SOC, we can conclude that the SOC should have small effect on the thermoelectric performance of $\text{Pb}_7\text{Bi}_4\text{Se}_{13}$.

The following statement is appended into the main text (Page 26).

The SOC effect on phonon dispersion and phonon DOS has also been considered, which demonstrates a limited effect on the calculated phonon spectra (See Figure S14).

Comment 4: What are concrete ways in experiment to tune the bandgap and increase the quality factor in the considered materials?

Response: Thanks for this question. The following discussion is added into the main text (Page 19-20).

For the tuning of bandgap, the chemical substitution of selenium by sulfur or tellurium tends to enlarge and decrease band gaps, respectively. To increase the quality factor, the introduction of coherent nanoscale defects is helpful, which could guarantee the unchanged carrier mobility, concomitantly strengthen defect-phonon scattering and further decreases lattice thermal conductivity. This could be realized chemically by introducing nanoscale precipitate, SrSe into $\text{Pb}_7\text{Bi}_4\text{Se}_{13}$ host matrix. This chemical strategy is used in the typical PbTe thermoelectric materials (*Nature Chemistry*, 2011, 3, 160-166.; *Advanced Energy Materials*, 2012, 2, 1117-1123.). There are still a variety of chemical methods to increase the quality, such as decreasing band offsets to increase the effective band convergence by doping heterogeneous elements in Pb or Bi sites.

Response to the reviewer #2:

In this manuscript, the authors report on a joint experimental and theoretical investigation of the thermoelectric properties of the layered compound $\text{Pb}_7\text{Bi}_4\text{Se}_{13}$ doped with various elements on the Pb sites. A central aspect of the manuscript is the discovery of high dimensionless thermoelectric figure of merit ZT ascribed to the band convergence of several conduction bands driven by the dopants introduced. The experimental and theoretical results are sound and certainly deserve to be published somewhere. However, I do not find that this manuscript provides a sufficient level of novel insights or thought-provoking ideas to be publishable in Nature Communications. As mentioned by the authors in their introduction, related compounds in various similar homologous series have been studied for their thermoelectric properties. In addition, the beneficial role played by the convergence of several electronic bands on the thermoelectric properties have been widely discussed already in the literature. Adding another example of such behaviour is certainly interesting but appears insufficiently novel. I think the present manuscript would be a better fit for Communication Materials or Scientific Report, provided that the following remarks/issues have been carefully addressed by the authors.

We thank for the suggestions and comments from reviewer #2. It is admitted that the $\text{Pb}_7\text{Bi}_4\text{Se}_{13}$ and compounds with similar structures was reported before. And the band convergence is indeed in favor of thermoelectric energy conversion efficiency. However, the record high zT value realized in this work is not only from the band convergence. Instead, the band nestification equally plays a significant role. In thermoelectric community, most work emphasizes on the effect of band convergence, which requires electronic bands at different k points in the Brillouin zone (BZ). The band nestification is rarely documented and frequently neglected. Here we show it is significant for the superior thermoelectric performance. More importantly, here we develop a new concept of *unique quality factor*. “The concept is developed in this paper for addressing both band non-parabolicity and bipolar effect, which are very difficult for the earlier definition of quality factor (*B Factor*) to evaluate”, as mentioned by reviewer 3. This new quality provides a time- and labor-saving method to evaluate and predict thermoelectric materials without expensive calculations on electron-phonon coupling at high temperatures.

Comment 1: This material belongs to the larger groups of homologous series based on Pb, Bi and Se, which have been already (at least for some of them) investigated for their interesting thermoelectric properties. In my opinion, the list of references given by the authors lack numerous relevant papers on these compounds. In addition, the authors may put into a broader perspective these materials by mentioning that other series, structurally related to lillianites, also exist (see for instance the following reference and the references therein: Lu et al. Mater. Adv. DOI: 10.1039/d0ma00912a (2021)).

Response: Thanks for this suggestion. Since $\text{Pb}_7\text{Bi}_4\text{Se}_{13}$ is a lillianite-type compounds, so we list the lillianite homogeneous series and some analogous to compare. According to this suggestion, the majority of compounds with similar structures have been enumerated here. As mentioned by reviewer #2 in this newly published work, Lu et al. Mater. Adv. DOI: 10.1039/d0ma00912a (2021), the compound $\text{Sn}_4\text{Bi}_{10}\text{Se}_{19}$ with the highest zT of 0.2 at 535 K is also included.

Figure S2. zT value and lattice thermal conductivity, κ_L of lillianite, tetradymite, pavonite, cannizzarite, galenobismuthite, complex rare-earth sulfides and other structure-similar compounds

After searching for compounds with similar lillianite-type structures, 37 different compounds (including $\text{Pb}_7\text{Bi}_4\text{Se}_{13}$ here) are found and compared here. As shown in Figure S2, the zT value and lattice thermal conductivity, κ_L of lillianite (red circle), tetradymite (half gray circle), pavonite (half green star), cannizzarite (half triangle), galenobismuthite (half pentagon), complex rare-earth sulfides (cross circle) and other structure-similar other compounds (half cyan square) are presented. It is obvious that the $\text{Pb}_7\text{Bi}_4\text{Se}_{13}$ marked by the blue circle not only demonstrates the highest zT value

among lillianite type structures, but also exhibits the competitive thermoelectric performance, compared to other compounds with similar structures. This comparison highlights the promise of $\text{Pb}_7\text{Bi}_4\text{Se}_{13}$ as an efficient candidate for thermoelectric applications. These detailed parameters of compounds shown in Figure S2 is also tabulated in Table S2.

The following discussion is appended into the main text (Page 9-10).

It is worth noting that this is the highest zT achieved to date not only in lillianite homogeneous series, but also in Tetradymite, Pavonite, Cannizzarite, Galenobismuthite, complex rare-earth sulfides and other structure-similar compounds. The total number of 37 different compounds were enumerated and compared in Figure S2, which shows the competitive performance of $\text{Pb}_7\text{Bi}_4\text{Se}_{13}$. The detailed lattice thermal conductivity and zT values of 37 distinct compounds are tabulated in Table S2.

This table is added into the supplementary information (Page S4-6).

TableS2. Lattice thermal conductivities and zT values of compounds with lillianite-similar structures

Compound	κ_L (W/mK)	zT	T (K)	Category	Literature
$\text{Pb}_3\text{Bi}_2\text{S}_6$	0.57	0.26	715	Lillianite	1
$\text{Ag}_2\text{Pb}_6\text{Bi}_{10}\text{Se}_{22}$	0.28	0.23	523		2
$\text{Pb}_6\text{Bi}_2\text{Se}_9$	0.95	0.25	673		3
$\text{SnPb}_2\text{Bi}_2\text{S}_6$	0.69	0.3	770		4
$\text{Pb}_5\text{Bi}_{12}\text{Se}_{23}$	0.42	0.25	723		5
$\text{Pb}_5\text{Bi}_{18}\text{Se}_{32}$	0.48	0.2	723		5
$\text{Sn}_4\text{Bi}_{10}\text{Se}_{19}$	0.26	0.2	535		6
$\text{Sn}_4\text{Bi}_2\text{Se}_7$	0.92	0.04	450		7
$\text{Sn}_2\text{Bi}_2\text{Se}_5$	0.69	0.08	525		7
SnBi_4Se_7	0.93	0.2	673		7
[*] $\text{Sn}_2\text{Pb}_5\text{Bi}_4\text{Se}_{13}$	--	--	300		8
[*] $\text{Sn}_{8.65}\text{Pb}_{0.35}\text{Bi}_4\text{Se}_{15}$	--	--	300		8
$\text{K}_x\text{Sn}_{6+2x}\text{Bi}_{2+2x}\text{Se}_9$	--	--	--		9
$\text{KSn}_5\text{Bi}_5\text{Se}_{13}$	--	--	--		9
[§] $\text{LiPbSb}_3\text{S}_6$	0.21	--	723		10
PbBi_2Te_4	0.76	0.4	650	Tetradymite	11
PbBi_4Te_7	0.6	0.5	600		11
PbSb_2Te_4	0.48	0.14	340		12

$\text{CdSnBi}_4\text{Se}_8$	0.44	0.4	850	Pavonite	13
$\text{CdPbBi}_4\text{Se}_8$	0.28	0.63	850		13
$\text{Cu}_{1.61}\text{Bi}_{4.81}\text{S}_8$	0.33	0.2	675		14
$\text{CdPb}_2\text{Bi}_4\text{S}_9$	0.73	0.53	775		15
$\text{CdAg}_2\text{Bi}_6\text{Se}_{11}$	0.35	0.95	775		15
$\text{LiSn}_2\text{Bi}_5\text{S}_{10}$	0.5	0.54	825		16
* $\text{NaSn}_2\text{Bi}_5\text{S}_{10}$	--	--	--		16
$\text{Pb}_5\text{Bi}_6\text{Se}_{14}$	0.29	0.46	705	Cannizzarite	17
$\text{Pb}_5\text{Bi}_6\text{Se}_{14-x}\text{I}_x$	0.3	0.5	723		17
PbBi_2S_4	0.48	0.33	710	Galenobismuthite	1
$(\text{GdS})_{1.20}\text{NbS}_2$	1.5	0.09	873	Complex rare-earth sulfides	18
$(\text{Gd}_{0.5}\text{Dy}_{0.5}\text{S})_{1.21}\text{NbS}_2$	1.1	0.13	873		18
$(\text{DyS})_{1.22}\text{NbS}_2$	1.1	0.12	873		18
$(\text{GdS})_{0.60}\text{NbS}_2$	1.9	0.03	873		18
$(\text{LaS})_{1.14}\text{NbS}_2$	0.9	0.15	950		18
$(\text{Yb}_2\text{S}_2)_{0.62}\text{NbS}_2$	0.4	0.1	300		18
MnSb_2Se_4	1.35	0.01	300	Others	19
$\text{MnPb}_{16}\text{Sb}_{14}\text{S}_{38}$	0.227	0.018	725		20
$\text{Pb}_7\text{Bi}_4\text{Se}_{13}$	0.17	1.35	800	Lillianite	This work

‡ For $\text{Sn}_2\text{Pb}_3\text{Bi}_4\text{Se}_{13}$ and $\text{Sn}_{8.65}\text{Pb}_{0.35}\text{Bi}_4\text{Se}_{15}$, only the electrical properties were measured and reported. In details, their electrical conductivities were measured from around 25 K to 300 K, while their Seebeck coefficients were collected from around 300 K to 675 K. No thermal conductivities are available. Here only the room temperature electrical conductivities, about 14 S/cm and 10 S/cm for $\text{Sn}_2\text{Pb}_3\text{Bi}_4\text{Se}_{13}$ and $\text{Sn}_{8.65}\text{Pb}_{0.35}\text{Bi}_4\text{Se}_{15}$ are extracted. And -200 $\mu\text{V/K}$ and 130 $\mu\text{V/K}$ for $\text{Sn}_2\text{Pb}_3\text{Bi}_4\text{Se}_{13}$ and $\text{Sn}_{8.65}\text{Pb}_{0.35}\text{Bi}_4\text{Se}_{15}$ are obtained.

‖ For $\text{K}_x\text{Sn}_{62x}\text{Bi}_{2+x}\text{Se}_9$ and $\text{KSn}_3\text{Bi}_5\text{Se}_{13}$, the crystal structure data was report without the measurement of electrical and thermal properties.

§ For $\text{LiPbSb}_3\text{S}_6$, only crystal structure and thermal conductivity are available.

* For $\text{NaSn}_2\text{Bi}_5\text{S}_{10}$, its band gap is 0.07 eV, whose electrical and thermal properties are not measured.

Comment 2: One major technical issue in this manuscript is the lack of measurements performed parallel and perpendicular to the pressing direction. This compound crystallizes with a monoclinic crystal structure (a drawing of the crystal structure should be shown in the manuscript) and, as widely shown in the literature, polycrystalline samples exhibit significant anisotropy between these two directions. Thus, the lack of such measurement and discussion is unacceptable and not serious. Unfortunately, I believe that the high ZT values reported in this manuscript are enhanced due to an erroneous combination of measurements in two different direction for the electrical and thermal properties. The authors should therefore show measurements performed in both directions (both Hall effect and transport properties) to strengthen their claim of high ZT values achieved.

Response: Thanks for this comment. This anisotropy of thermoelectric properties of $\text{Pb}_7\text{Bi}_4\text{Se}_{13}$ has already been considered in this work. As shown in the previous Figure S4 (now it is updated to Figure S3) in the supplementary information, we have

already measured electrical and thermal properties of two different samples, parallel and perpendicular to the pressing direction. In the main text, the thermoelectric properties of these doped $\text{Pb}_7\text{Bi}_4\text{Se}_{13}$ are properties measured parallel to the pressing direction. We want to emphasize here that our zT value calculation are consistent from the same direction of electrical conductivity, Seebeck coefficient, and thermal conductivity.

To clarify the anisotropy of thermoelectric properties, a schematic illustration on measurement geometry, Hall measurement of two different compositions along two directions have been added into the supplementary information. In addition, the thermoelectric properties of $\text{Pb}_7\text{Bi}_4\text{Se}_{13}$ with the optimal performance is also measured parallel and perpendicular to the pressing direction.

The following statement is appended into the main text (Page 8).

It should be noted that there exists a small degree of anisotropy of thermoelectric properties, due to the monoclinic structure of $\text{Pb}_7\text{Bi}_4\text{Se}_{13}$. The measurement geometry and thermoelectric properties parallel and perpendicular to the SPS (spark plasma sintering) directions of two different samples are presented in Figure S3. In this work, only the thermoelectric properties parallel to the SPS direction is presented.

The following experimental results are presented into the supplementary information (Page S7-9).

Figure S3. Thermoelectric properties of $\text{Pb}_7\text{Bi}_4\text{Se}_{13}$ with $n_H = 9.2 \times 10^{20}$ (Sample 1) and 7.4×10^{20} cm^{-3} (Sample 2) parallel (\parallel) and perpendicular (\perp) to SPS direction. (a) Measurement geometry, (b) Seebeck coefficient, (c) electrical conductivity, (d) power factor (PF), (e) total thermal conductivity and lattice thermal conductivity, (f) zT values, (g) electrical conductivity at low temperature, (h) Hall carrier concentration, (i) mobility. The thermoelectric properties are measured in ZEM device and the Hall measurement is completed in PPMS instrument.

As shown in Figure S3, the measurement geometry shows the electrical and thermal conductivities measured parallel and perpendicular to the SPS direction. The Seebeck coefficient, electrical conductivities, thermal conductivity and zT values of two different compositions were measured from RT to 800 K, as shown in Figure S3b-3f. Electrical conductivities, Hall carrier concentration and mobility were also measured and presented in Figure S3g-3i. It is clear that there exists a very small degree of anisotropy of thermoelectric properties, which originates from the non-cubic crystal structure of $\text{Pb}_7\text{Bi}_4\text{Se}_{13}$. Meanwhile, the PF and thermal conductivities in two different measurement directions are close, as shown in Figure 3d-3e, which leads to the similar zT values in the two directions.

Figure S4. Thermoelectric properties of $\text{Pb}_7\text{Bi}_4\text{Se}_{13}$ with $n_H = 1.2 \times 10^{20} \text{ cm}^{-3}$ parallel (\parallel) and perpendicular (\perp) to the SPS direction. (a) Seebeck coefficient, (b) electrical conductivity, (c) power factor (PF), (d) total thermal conductivity and lattice thermal conductivity, (e) electronic thermal conductivity, (f) zT values.

To clarify the anisotropy of thermoelectric properties of $\text{Pb}_7\text{Bi}_4\text{Se}_{13}$ with $n_H = 1.2 \times 10^{20}$, we present the Seebeck coefficient, electrical conductivities, thermal conductivities and zT values measured parallel and perpendicular to the SPS direction. The measurement geometry is presented in Figure S3a. Clearly, there exist a very small degree of anisotropy of thermoelectric properties as shown in Figure S4a-4c, due to the non-cubic crystal structure of $\text{Pb}_7\text{Bi}_4\text{Se}_{13}$. As shown in Figure S4d, the thermoelectric performance parallel to the SPS direction is slightly better than that perpendicular to the SPS direction. Consequently, thermoelectric properties of doped $\text{Pb}_7\text{Bi}_4\text{Se}_{13}$ in the main text are results parallel to the SPS direction.

In this paragraph, a detailed discussion on the anisotropy of thermoelectric properties is performed. Thermoelectric materials with non-cubic structures demonstrate different degrees of anisotropy of charge and phonon transport. The anisotropy is extremely large in some layered structures, such as Bi_2Se_3 , BiSe and

their derivatives. $\text{Pb}_5\text{Bi}_6\text{Se}_{14}$, a member of cannizzarite-type structure, also exhibits the different electrical and thermal conductivities in parallel and perpendicular directions.¹⁷ It mainly originates from its structural features, in which PbSe and Bi_2Se_3 subunit stack alternatively along the c axis. This layered structure is easily to form preferred orientation in the pressing direction. By contrast, other compounds, such as $\text{Pb}_3\text{Bi}_2\text{S}_6$,¹ $\text{SnPb}_2\text{Bi}_2\text{S}_6$ (members of lillianite homologous series),⁴ and PbBi_2S_4 (a member of galenobismuthite homologous series),¹ demonstrate nearly isotropic electrical and thermal transport properties. Taking $\text{Pb}_3\text{Bi}_2\text{S}_6$ as an instance, it consists of NaCl-type (Pb/Bi)S layers with a mirror as twinning operation. These layers form the NaCl-type strips and avoid the formation of single layered structure preferred stacking to any crystal axis. Under pressing, crystal grains tend to distribute randomly, which gives rise to the near isotropy in electrical and thermal properties of parallel and perpendicular directions. For $\text{Pb}_7\text{Bi}_4\text{Se}_{13}$, its crystal structure includes the NaCl-type (Pb/Bi)Se strips without separate PbSe and BiSe layer stacking alternatively. This might give rise to this anisotropy of thermoelectric properties. In total, the anisotropy of electrical and thermal transport properties is structural dependence. Nearly isotropic electrical and thermal transport properties are also observed in $\text{Pb}_3\text{Bi}_2\text{S}_6$, $\text{SnPb}_2\text{Bi}_2\text{S}_6$, and PbBi_2S_4 . $\text{Pb}_7\text{Bi}_4\text{Se}_{13}$ only demonstrates a certain degree of anisotropy in electrical and thermal conductivities.

As regard to crystal structure of $\text{Pb}_7\text{Bi}_4\text{Se}_{13}$, it has already been presented into the Figure S1a. And the Figure 5d and 5f in the main text also include the information of crystal structure with atomic displacements. For the conciseness of this work, the crystal structure information seems to be enough.

Comment 3: In Figure 3, the authors show the Pisarenko plot together with the theoretical line calculated using a single-non-parabolic band model. In principle, for non-parabolic bands, the density of states effective mass (m^*) should vary with the carrier concentration. In the present case, even spanning one order of magnitude does not result in significant variations in m^* . May the authors explain in detail the reasons?

The authors may show a theoretical line calculated with a single-parabolic band model to strengthen their claim. In addition, the values of K in equations (1) and (2) used for these calculations should be mentioned as well as the way they have been determined. Do the results presented depend on the K values? The same questions apply for the inertial mass. As a minor point, the $(n,m,k)F$ functions are not the Fermi integrals (that appear in the SPB model) but rather the generalized Fermi integrals.

Response: Thanks for this question. The Pisarenko plot by using single parabolic band (SPB) model with the same effective mass is also added into Figure 3. In addition, the Pasarenko plots of a series of effective masses by SPB and SKB models are presented in FigureS10. It is clearly that both SPB and SKB could accurately describe the Seebeck coefficient with the same effective mass in Figure S10a. However, the temperature dependence of effective mass increases with increasing temperature, which shows the feature of Kane band (Figure S10b). This feature is commonly observed in PbTe with typical non-parabolic bands.^{21, 22}

Figure S10. (a) Hall carrier concentration as a function of Seebeck coefficient (Pisarenko plot) by using single parabolic band (SPB) and single Kane band (SKB) models with different effective mass. (b) Temperature dependence of effective mass of two compositions, Ga1 and Ga2.

The Figure S10 and the following statement are appended into the supplementary information (Page S14).

By using a series of effective mass, m_s^* , Hall carrier concentration as a function of Seebeck coefficient (Pisarenko plot) at 300 K is presented in Figure S10a. The SPB and SKB models are established based on the assumption of the dominant acoustic

phonon scattering (APS). For the prediction of Seebeck coefficient, both SPB and SKB could equally work with the same effective mass, also observed in $\text{PbTe}_{1-x}\text{I}_x$.²² Actually, the m_s^* of $\text{Pb}_7\text{Bi}_4\text{Se}_{13}$ compounds are increasing with increasing temperature, consistent with Kane band feature. This temperature dependence of m_s^* is commonly observed in PbTe ,²¹ in which the SKB model is utilized to describe its Seebeck coefficient and mobility.

With regard to the unchanged effective mass observed in the Pisarenko plot in the Figure 3c. The following discussion is extended here.

Different from the parabolic band, the effective mass is not well defined in Kane band due to different definitions for effective mass, m^* , based on distinct classical relationships. According to the non-parabolic band $E \sim k$ dispersion, the widely accepted energy-dependent effective mass derived from the electron momentum is:

$$E\left(1 + \frac{E}{E_g}\right) = \frac{\hbar^2 k^2}{2m^*},$$

This definition is found in most solid-state physics textbook, $m^* = \hbar(d^2 E/dk^2)^{-1}$, which directly relates the m^* to the band curvature. And this $E \sim k$ dispersion is obtained in many different measurements, such as optical reflectance, Shubnikov-De Haas/De Haas-Van Alphen oscillations, Faraday rotation and combined galvanomagnetic measurements. In general, the effective mass derived from these methods increases with increasing energy. However, such an increasing trend with energy could not be expected on the Seebeck Pisarenko plot. In thermoelectric community, we define the Seebeck effective mass, m_s^* as the density of state (DOS) effective mass, which predict the Seebeck coefficient with Hall carrier concentration, n_H in the SPB and SKB models. Consequently, it should be realized that the qualitative difference between m_s^* (Seebeck effective mass) and m^* (effective mass from the $E \sim k$ dispersion). This Seebeck effective mass, m_s^* is defined by the following equations:

$$S = \frac{k_B}{e} \left(\frac{{}^1 F_{-2}^1}{{}^0 F_{-2}^1} - \eta \right)$$

$$n_H = \frac{(2m_s^* k_B T)^{3/2}}{3\pi^2 \hbar^3} \cdot \frac{(2K+1)^2}{3K(K+2)} \cdot \frac{({}^0 F_{-2}^1)^2}{{}^0 F_{-4}^{1/2}}$$

The Seebeck effective mass derived from the Pisarenko plot does not show obvious changes with increasing energy. Actually, this unchanged Seebeck effective

mass in SKB model has been commonly observed in previous works, which also cited here in Figure R1 for clarifying this question. In Figure R1, the effective mass derived from the Pisarenko plot based on SKB still remains unchanged even the carrier concentration increases from below 10^{19} cm^{-3} to 10^{20} cm^{-3} .

[Redacted]

Figure R1. Pisarenko plot of carrier concentration as a function Seebeck coefficient of (a) $\text{PbTe}_{1-x}\text{I}_x$ (Aaron LaLonde, *et al*, *Energy Environ. Sci.* 2011, 4, 2090.) (b) La- and I-doped PbTe (Yanzhong Pei, *et al*, *Energy. Environ. Sci.* 2012, 5, 7963.)

And this discussion is also appended into the main text (Page 15).

The temperature dependence of effective mass verifies its band nonparabolicity, as shown in Figure S10b. It should be noted that the Seebeck effective mass, m_s^* is defined as the density of state effective mass in thermoelectric community, which predicts the Seebeck coefficient with Hall carrier concentration, n_H in both the SPB and SKB models. However, it should be also realized that the qualitative difference between m_s^* (Seebeck effective mass) and m^* (effective mass from the $E \sim k$ dispersion). The Seebeck effective mass is defined by equations (1) and (2), which remains unchanged with increasing n_H . Differently, the effective mass, m^* defined by $m^* = \hbar (d^2 E/dk^2)^{-1}$, increases with increasing E .

As regard to the K , the following statement is also added into the main text (Page 16).

The K in equations (1) and (2) of the main text indicates the anisotropy of Fermi surface. It is defined by $K = m_{\parallel}^*/m_{\perp}^*$, in which the m_{\parallel}^* and m_{\perp}^* exhibit the longitudinal and transverse effective mass. This value of K could be obtained by DFT calculation. In most cases, it is assumed to be 1. The different K should affect the values of effective mass.

Comment 4: The quality factor calculated for non-parabolic bands has already been done for

PbTe-based compounds. The authors should discuss similarities/differences between these well-known thermoelectric compounds and the present lillianite sample.

Response: Thanks for this suggestion. The quality factor with single Kane band (SKB) has been established for PbTe-based compounds. Since PbTe-based compounds exhibit narrow band gaps, the bipolar conduction plays a critical role at high temperature, especially for T above 600 K. Consequently, this previously established quality factor without considering bipolar effect can only be used at low temperatures, such as La- or I-doped PbTe at room temperature or mediate temperature (600 K). Until now, there is no report on the quality factor with two Kane band (TKB) model. To evaluate thermoelectric performances at high temperature, it is essential and critical to include the bipolar effect. In this work, we establish this unique quality factor by using the TKB model, which could simultaneously consider the band nonparabolicity and bipolar effect. This unique quality factor provides a more precise approach to evaluate performances of thermoelectric materials with narrow band gaps. The concept of the *unique quality factor* is developed in this paper for addressing both band non-parabolicity and bipolar effect, which are very difficult for the earlier definition of quality factor (B Factor) to evaluate.

The is following statement is the discussion on the difference and similarities on typical thermoelectric materials and $\text{Pb}_7\text{Bi}_4\text{Se}_{13}$, which is appended into the main text (the discussion section, Page 29).

For $\text{Pb}_7\text{Bi}_4\text{Se}_{13}$, conduction bands are highly nested at Y_2 and M_2 points. Besides, these nested conduction bands at Y_2 and M_2 also share similar energies, demonstrating a strong signature of band convergence. The synergistic effect of band nestification and convergence leads to higher band degeneracy and superior electrical properties. This electronic band features are rarely documented in compounds with lower crystal symmetries, and are comparable to typical n -type thermoelectric materials, such as PbTe and CoSb_3 , which both demonstrates multiple degenerated bands as well as narrow band gaps ($E_g = \sim 0.3$ eV for PbTe and ~ 0.23 eV for CoSb_3). For PbTe, conduction bands consist of one Kane band at L point. The full valley

degeneracy, N_v for L are 4. For CoSb_3 , conduction bands are composed of one Kane band at Γ point ($N_v = 3$) and one parabolic band along Γ -N line ($N_v = 12$). Moreover, $\text{Pb}_7\text{Bi}_4\text{Se}_{13}$ demonstrates acoustic phonons with large Grüneisen parameters and low-frequency optical phonons, which facilitates strong phonon scattering and ultralow lattice thermal conductivity.

The following discussion is on the comparison of quality factors established on SPB, SKB and TKB models. And the discussion is added into the main text (Page 20-21).

It should be noted that the quality factor with SKB model has already been established, which could evaluate and predict the thermoelectric performance. However, its application is limited at low temperatures, due to thermally excited minor carrier with increasing temperature. The contribution of minor carrier becomes inevitable and considerably deteriorates thermoelectric performances. A lack of effective quality factor brings a great challenge to evaluate and predict thermoelectric performances in semiconductors with narrow band gaps, especially at high temperatures. To resolve this issue, this unique quality factor is established with TKB model, which can provide an effective and time-saving method. The derivations on previously reported quality factors established on SPB and SKB models (called B_{Para} and B_{Kane} for clarity), and newly developed B_{Kane}^* are presented in the section 5 of supplementary information. With regard to previously reported quality factors, zT values only correlates with the B_{Para} (or B_{Kane}) and reduced chemical potential, η . By contrast, the zT in this unique quality factor depends on three independent variables, B_{Kane}^* , η and ξ . The 3D and contour plots of three different quality factors are presented in Figure S12.

The following discussion focuses on details of quality factors with SPB, SKB, and TKB models and added into the section 5 of the supplementary information (Page S22-25).

1. Quality factor by single parabolic band model

$$\begin{aligned}
zT &= \frac{S^2 \sigma T}{\kappa_L + \kappa_e} \\
&= \frac{S^2}{\frac{\kappa_L}{\sigma T} + L} \\
&= \frac{S^2}{\frac{(k_B/e)^2}{(\frac{k_B}{e})^2 \cdot \frac{\sigma_{E_0}}{\kappa_L} T \cdot F_0} + L} \\
&= \frac{S^2}{\frac{(k_B/e)^2}{B_{Para} \cdot F_0} + L} \\
zT &= \frac{S^2 \sigma T}{\kappa_L + \kappa_e} = \frac{S^2}{\frac{(k_B/e)^2}{B_{Para} \cdot F_0} + L}
\end{aligned}$$

Quality factor, $B_{Para} = \left(\frac{k_B}{e}\right)^2 \frac{\sigma_{E_0}}{\kappa_L} T$

Transport coefficient, $\sigma_{E_0} = \frac{\sigma}{F_0} = \frac{2\hbar e^2}{3\pi} \cdot \frac{N_V v_l^2 d}{m_l^* \Xi^2}$

2. Quality factor by single Kane band model

$$\begin{aligned}
zT &= \frac{S^2 \sigma T}{\kappa_L + \kappa_e} \\
&= \frac{S^2}{\frac{\kappa_L}{\sigma T} + L} \\
&= \frac{S^2}{\frac{(k_B/e)^2}{(\frac{k_B}{e})^2 \cdot \frac{\sigma_{E_0}}{\kappa_L} T \cdot 3^0 F_{-2}^1} + L} \\
&= \frac{S^2}{\frac{(k_B/e)^2}{B_{Kane} \cdot 3^0 F_{-2}^1} + L} \\
zT &= \frac{S^2 \sigma T}{\kappa_L + \kappa_e} = \frac{S^2}{\frac{(k_B/e)^2}{B_{Kane} \cdot 3^0 F_{-2}^1} + L}
\end{aligned}$$

Quality factor, $B_{Kane} = \left(\frac{k_B}{e}\right)^2 \frac{\sigma_{E_0}}{\kappa_L} T$

Transport coefficient, $\sigma_{E_0} = \frac{\sigma}{3 \cdot {}^0 F_{-2}^1} = \frac{2\hbar e^2}{3\pi} \cdot \frac{N_V v_l^2 d}{m_l^* \Xi^2}$

3. Quality factor by two Kane band model

$$\begin{aligned}
zT &= \frac{S^2 \sigma T}{\kappa_L + \kappa_c + \kappa_b} \\
&= \frac{S^2}{\frac{\kappa_L}{\sigma T} + \frac{\kappa_c}{\sigma T} + \frac{\kappa_b}{\sigma T}} \\
&= \frac{\left(\frac{S_e \gamma + S_h}{\gamma + 1}\right)^2}{\frac{\gamma}{\gamma + 1} \cdot \frac{\left(\frac{k_B}{e}\right)^2}{3B_{Kane}^* \cdot {}^0F_{-2}^1} \cdot \xi + \frac{L_e \gamma + L_h}{\gamma + 1} + \frac{\gamma}{(\gamma + 1)^2} (S_e - S_h)^2} \\
&= \frac{(S_e \gamma + S_h)^2}{\gamma(\gamma + 1) \cdot \frac{\left(\frac{k_B}{e}\right)^2}{3B_{Kane}^* \cdot {}^0F_{-2}^1} \cdot \xi + (\gamma + 1)(L_e \gamma + L_h) + \gamma(S_e - S_h)^2} \\
&= \frac{(S_e \gamma + S_h)^2}{(\gamma + 1) \left[\frac{\left(\frac{k_B}{e}\right)^2 \cdot \gamma \xi}{3B_{Kane}^* \cdot {}^0F_{-2}^1} + (L_e \gamma + L_h) \right] + \gamma(S_e - S_h)^2} \\
zT &= \frac{S^2 \sigma T}{\kappa_L + \kappa_c + \kappa_b} = \frac{(S_e \gamma + S_h)^2}{(\gamma + 1) \left[\frac{\left(\frac{k_B}{e}\right)^2 \cdot \gamma \xi}{3B_{Kane}^* \cdot {}^0F_{-2}^1} + (L_e \gamma + L_h) \right] + \gamma(S_e - S_h)^2}
\end{aligned}$$

Quality factor, $B_{Kane}^* = \frac{k_B}{e^2} \cdot \frac{\sigma_{E_0} E_g}{\kappa_L}$

Transport coefficient, $\sigma_{E_0} = \frac{\sigma}{3 \cdot {}^0F_{-2}^1} = \frac{2\hbar e^2}{3\pi} \cdot \frac{N_V v_i^2 d}{m_l^* \Xi^2}$

FigureS12. (a) 3D plot of figure of merit, zT as functions of reduced Fermi level, η and quality factor, B_{Para} . (b) Contour plot of η - and B_{Para} -dependent zT . The white dash lines show a series of zT values from 0.2 to 1.2. This B_{Para} is established on the single parabolic band model (SPB). (c) 3D plot of figure of merit, zT as functions of reduced Fermi level, η and quality factor, B_{Kane} . (d) Contour plot of η - and B_{Kane} -dependent zT . The white dash lines show a series of zT values from 0.2 to 1.4. This B_{Kane} is established on the single Kane band model (SKB). (e) 3D plot of figure of merit, zT as functions of reduced Fermi level, η and reduced band gap, ξ with quality factor, $B_{Kane}^* = 5$. (f) Contour plot of η - and ξ -dependent zT . The white dash lines show a series of zT values from 0.2 to 1.2. This B_{Kane} is established on the two Kane band model (TKB).

Figure S12 shows the different quality factors established on SPB, SKB, and TKB models, respectively. The quality factor with SPB model is widely used in thermoelectric materials with wide band gaps, such as PbSe, SnSe, BiCuSeO, *et al.* For thermoelectric materials with narrow bandgaps, the band nonparabolicity is

inevitable throughout the entire temperature ranges. The quality factor with single Kane band model is utilized at room temperature or moderate temperature, such as La- and I-doped PbTe. Until now, there is no report on quality factor both considering band nonparabolicity and bipolar effect, which greatly limit the precise evaluation and prediction of thermoelectric materials with narrow band gaps, such as PbTe, Bi₂Se₃, CoSb₃, SnTe, *et al* and unconventional Pb₇Bi₄Se₁₃ here. Here, the unique quality factor, B^*_{Kane} , established on the TKB model, considers the band nonparabolicity and bipolar effect, which leads to more accurate and reliable results.

Comment 5: Several typos are present in the manuscript and should be corrected, for instance “Lorentz” should be “Lorenz”, etc....

Response: Thanks. This manuscript and supplementary information have been updated to correct these typos.

Comment 6: Regarding the lattice dynamics of this compound, a rapid survey of the literature indicates that an experimental study of the lattice dynamics of the (PbSe)₅(Bi₂Se₃)_{3m} compounds has been published (Sassi et al., Phys. Chem. Chem. Phys. 20, 14597 (2018)). The authors should mention at minima this reference and discuss/compare their results with those obtained on these related materials.

Response: Thanks for this suggestion. The discussion and comparison on the lattice dynamics of this work and (PbSe)₅(Bi₂Se₃)_{3m} have been added into this manuscript. The related reference literatures have been cited.

The following statement is appended into the main text (Page 24-25).

Actually, the low-energy vibrational modes in phonon DOS have been directly observed in (PbSe)₅(Bi₂Se₃)_{3m} by the inelastic neutron scattering measurement, which shares similar structural complexity and compositions with Pb₇Bi₄Se₁₃. Besides, the maxima centered below 10 K in the relationship of $T \sim C_p/T^3$ have also been observed in (PbSe)₅(Bi₂Se₃)_{3m}. These features have also been reported in compounds with low thermal conductivities, such as BaGa₅ and InTe.

Response to the reviewer #3:

In the introduction, the authors present their hypothesis for bringing up the thermoelectric performance of doped $\text{Pb}_7\text{Bi}_4\text{Se}_{13}$ to an exceptional level. They list favorable parameters that the material exhibits, such as bond anharmonicity, which they later quantify through phonon calculations, and the existence of lone electronic pairs, all reducing the thermal conductivity.

To overcome the limitations of poor electrical conductivity in $\text{Pb}_7\text{Bi}_4\text{Se}_{13}$, they dope the material with Ga, In, Ag, and I, searching for the best dopant candidate. Out of the dopants, Ga in the composition $(\text{Pb}_{0.95}\text{Ga}_{0.05})_7\text{Bi}_4\text{Se}_{13}$ produces the best thermoelectric figure of merit of 1.35 at 800 K. The peak figure of merit tapers off very gently for lower temperatures. Through density functional calculations, they demonstrate that $\text{Pb}_7\text{Bi}_4\text{Si}_{13}$ has a flat valence band top and valley-degenerate conduction band bottom, both sought-after features in thermoelectric materials.

Moreover, the authors propose a new quality factor that considers the band non-parabolicity and the bipolar effect, as applicable to $\text{Pb}_7\text{Bi}_4\text{Se}_{13}$, in analyzing the figure of merit. This new quality factor predicts even potentially higher ZT for $\text{Pb}_7\text{Bi}_4\text{Se}_{13}$ that might be realized in the future.

In the last part of the paper, the authors, based on the TEM image analysis, identify various dislocations responsible for further reduction of the lattice conductivity.

This paper falls in the top tier of the thermoelectric research in the term of novelty and completeness. The results outperform the earlier reports on similar material (Ref. 43). Many similar research papers either lack complete mechanistic characterizations and discussions or do not present materials with a stellar performance. As a result, I think this paper can be published with a little more enhancement to the arguments presented. My recommended revisions are listed below:

1. What application do the authors think this material is best for? Industrial waste energy recovery at large scales? Or more minor applications such as in automobile parts?

Response: Thanks for this question. The urgent energy and environmental issues have a higher requirement on clean energy. Thermoelectric materials provide a promising opportunity to resolve the critical issue by scavenging waste heat into electricity. Considering the promising high zT of 1.35 at 800 K and the high average zT of 0.92 from 450 K to 800 K, the application of $\text{Pb}_7\text{Bi}_4\text{Se}_{13}$ is suitable for the middle-temperature power generation (500 K ~ 900 K), such as waste energy recovery, remote sensor powder and emergency power sources. In addition, it can be used in marine engine of ships by utilizing waste heat from exhaust pipes.

The following discussion is appended into the main text (Page 10).

This promising high peak and average zT values enable $\text{Pb}_7\text{Bi}_4\text{Se}_{13}$ to be applicable for middle-temperature power generation, such as waste energy recovery, remote sensor power and emergency power sources. Besides, it also can be used in marine engine of ships parts by utilizing waste heat from exhaust pipes.

2. Although $\text{Pb}_7\text{Bi}_4\text{Se}_{13}$ is thermoelectrically fully characterized, nothing was mentioned about the materials' thermomechanical properties. For instance, how robust are $\text{Pb}_7\text{Bi}_4\text{Se}_{13}$ and its doped variants towards successive cooling and heating cycles in terms of crack formation and detrimental ionic diffusion? (A more thorough literature search may provide some answers to the questions above that can be included in the discussion part.)

Response: Thanks for this comment. In addition to favorable thermoelectric energy conversion efficiency, robust thermomechanical properties are equivalently significant for thermoelectric materials. To investigate the thermal stability and mechanical response of $\text{Pb}_7\text{Bi}_4\text{Se}_{13}$, the following experiments are performed and the related discussion is added into the supplementary information (Page S11-12).

1. Compression test and nano-indentation investigation

Figure S6. The compressive strain–stress curves of $(\text{Pb}_{0.95}\text{Ga}_{0.05})_7\text{Bi}_4\text{Se}_{13}$. The strain rate is $5 \times 10^{-4} \text{ s}^{-1}$.

The result of compression test in Figure S6 indicates the strength and strain of $(\text{Pb}_{0.95}\text{Ga}_{0.05})_7\text{Bi}_4\text{Se}_{13}$. The UCS (ultimate compressive strength) of the $(\text{Pb}_{0.95}\text{Ga}_{0.05})_7\text{Bi}_4\text{Se}_{13}$ is about 85 MPa and the strain at fracture is about 1.2%.

Figure S7. Nanoindentation curves of $(\text{Pb}_{0.95}\text{Ga}_{0.05})_7\text{Bi}_4\text{Se}_{13}$. (a) Load/unload-depth curve, 33 cycles for one point. (b) Hardness and Elastic modulus at different depths. The standard error originates from averaging data from 5 different samples with the same composition.

The nanoindentation load/unload-depth curve is presented in Figure S7, which includes 33 cycles for one point. For the final result, 10 points for one sample and 5 samples have been measured and averaged. The hardness and elastic modulus at different depths is exhibited in Figure S7b. The hardness is estimated to be 2.88 ± 0.04 GPa, and the elastic modulus is 56 ± 4 GPa. Both the hardness and modulus reduce slightly with increasing the depth, due to size effect.

Figure S8. Microstructural investigation of thermal annealing of $(\text{Pb}_{0.95}\text{Ga}_{0.05})_7\text{Bi}_4\text{Se}_{13}$. (a) Photograph, (b) and (c) SEM images of ingot before annealing. (d) Photograph, (e) and (f) SEM images of ingot after annealing on 800 K in vacuum for 336 hours (2 weeks). The blue arrows indicate the ingot surface. The small yellow arrows mark the tiny cracks in the corners after thermal annealing.

To investigate the thermal stability of $(\text{Pb}_{0.95}\text{Ga}_{0.05})_7\text{Bi}_4\text{Se}_{13}$, this ingot was

subjected to the annealing at 800 K for 336 hours (2 weeks) in vacuum quartz tube. The microstructural features of the ingot before and after thermal annealing are compared. Compared to the ingot before annealing in Figure S8a, the surface of the ingot (indicated by the blue arrow) after annealing exhibits the features of completeness and smoothness, without pitting and bloating in Figure S8d. Only small cracks are observed in the ingot corners marked by the small yellow arrows. SEM images shows fracture surfaces before (Figure S8b-c) and after annealing (Figure S8e-f). It is obvious that the fracture surfaces after annealing share similar features with its counterpart, in which no obvious micropores can be detected. And the heating and cooling measurement (shown in Figure S5) shows a good repeatability. These investigation shows the $(\text{Pb}_{0.95}\text{Ga}_{0.05})_7\text{Bi}_4\text{Se}_{13}$ exhibits reasonably robust thermal stability.

The following discussion is added into the main text (Page10).

The compression test shows an ultimate compressive strength of 85 MPa and the strain can reach 1.2 % (Figure S6). Nano-indentation investigation indicates the hardness of 2.88 ± 0.04 GPa (Figure S7). And the microstructural investigation shows no obvious precipitates and micropores in the sample after annealing under vacuum condition for two weeks (Figure S8), which shows its reasonably robust thermal stability.

3. Furthermore, as I collect, the concept of the “unique quality factor” is developed in this paper for addressing both band non-parabolicity and bipolar effect, which are very difficult for the earlier definition of quality factor (B Factor) to evaluate. Since this is a new concept, can the authors demonstrate its applicability for different materials in the SI, or at least prove that it converges to B Factor if solved for a parabolic band?

Response: Thanks for this suggestion. The previously defined quality factors (B) are established on the single parabolic band (SPB) model or single Kane band (SKB) model. Strictly, the SPB quality factor is suitable for thermoelectric materials with wide band gaps, such as $\text{Cu}_2\text{ZnGeSe}_4$ (~1.4 eV), Cu_{2-x}Se (~1.23 eV), BiCuSeO (~0.8 eV), SnSe (~0.86 eV), CsAgTe_5 (~0.67 eV), BiAgSeS (~0.65 eV), Ag_6GaSe_6 (~0.56 eV), Half-Heuslers alloys (such as TaFeSb 0.53 eV, NbFeSb 0.41 eV), PbS (~0.43 eV),

GeTe-AgSbTe₂ (~0.39 eV), *et al.* The SKB quality factor works for thermoelectric semiconductors with narrow band gaps, such as PbTe (~0.22 eV), PbSe (0.28eV), (Bi,Sb)₂Te₃ (~0.13 eV), BiSe (~0.13 eV), CoSb₃ (~0.23eV), *et al.* The SKB quality factor considers the band nonparabolicity, stemming from the interaction between conduction and valence bands. However, its application is severely limited at low temperatures, due to significant thermal excitation of minor carrier with increasing temperature. In this situation, the minor carrier contribution becomes inevitable and considerably deteriorates thermoelectric performances. There is no effective quality factor to evaluate and predict thermoelectric performances in semiconductors with narrow band gap, especially at high temperatures. To resolve this issue, unique quality factor is established with two Kane band model, which can provide an effective and time-saving method.

The following discussion is added into the main text (Page 20-21).

It should be noted that the quality factor with single Kane band has already been established, which could evaluate and predict the thermoelectric performance at low temperatures. However, its application is limited at low temperatures, due to thermally excited minor carrier with increasing temperature. The contribution of minor carrier becomes inevitable and considerably deteriorates thermoelectric performances. A lack of effective quality factor brings a great challenge to evaluate and predict thermoelectric performances in semiconductors with narrow band gap, especially at high temperatures. To resolve this issue, unique quality factor is established with two Kane band model, which can provide an effective and time-saving method. The derivations on the previously reported quality factor established on SPB and SKB models (called B_{Para} and B_{Kane} for clarity), and newly developed B_{Kane}^* are presented into the section 5 of the supplementary information. With regard to the previously reported quality factors, the zT values only correlates with the B_{Para} (or B_{Kane}) and reduced chemical potential, η . By contrast, the zT depends on three independent variables, B_{Kane}^* , η and ξ in this unique quality factor. The 3D and contour plots of three different quality factors are presented into the

Figure S12, which clearly shows their difference and similarities.

To show the difference among different quality factors, here we present the SPB quality factor (B_{Para}), SKB quality factor (B_{Kane}) and the unique quality factor (B_{Kane}^*). The detailed derivation on the three quality factors is added into the section 5 of supplementary information (Page S22-25).

Figure S12. (a) 3D plot of figure of merit, zT as functions of reduced Fermi level, η and quality factor, B_{Para} . (b) Contour plot of η - and B_{Para} -dependent zT . The white dash lines show a series of zT values from 0.2 to 1.2. This B_{Para} is established on the single parabolic band model (SPB). (c) 3D plot of figure of merit, zT as functions of reduced Fermi level, η and quality factor, B_{Kane} . (d) Contour plot of η - and B_{Kane} -dependent zT . The white dash lines show a series of zT values from

0.2 to 1.4. This B_{Kane} is established on the single Kane band model (SKB). (e) 3D plot of figure of merit, zT as functions of reduced Fermi level, η and reduced band gap, ξ with quality factor, $B_{Kane}^* = 5$. (f) Contour plot of η - and ξ -dependent zT . The white dash lines show a series of zT values from 0.2 to 1.2. This B_{Kane}^* is established on the two Kane band model (TKB).

This unique quality factor could work well in $\text{Pb}_7\text{Bi}_4\text{Se}_{13}$ as discussed in the main text. Here we will introduce its applications in different thermoelectric materials with narrow band gaps. The following discussion is added into the supplementary information (Page S26-27).

Figure S13. (a) 3D plot of figure of merit, zT as functions of reduced Fermi level, η and reduced band gap, ζ with quality factor, $B_{Kane}^* = 6$. (b) Contour plot of η - and ζ -dependent zT . The white dash lines show a series of zT values from 0.6 to 1.4. The white point comes from the experiment data of $(\text{Bi,Sb})_2\text{Se}_3$. The red point is the predicted highest zT value. (c) 3D plot of figure of merit, zT as functions of reduced Fermi level, η and reduced band gap, ζ with quality factor, $B_{Kane}^* = 0.24$. (d) Contour plot of η - and ζ -dependent zT . The white dash lines show a series of zT values from 0.10 to 0.25. This B_{Kane}^* is established on the two Kane band model (TKB).

To extend the application of this unique quality factor, we firstly apply it into the typical thermoelectric material, $(\text{Bi,Sb})_2\text{Te}_3$ with a narrow band gap of ~ 0.18 eV. Figure S13a-b show the 3D plot and contour plot of figure of merit, zT as functions of reduced Fermi level, η and reduced band gap, ζ in $(\text{Bi,Sb})_2\text{Te}_3$. The unique quality factor, B_{Kane}^* is calculated to be 6 by using physical parameters from *Pan et al.* The quality factor derivation method has been described in the corresponding sections of supplementary information. In Figure S13b, the experiment point is revealed by the

white point with the measured zT of ~ 0.76 . The highest predicted zT is 1.54, shown by the red point. The red arrow indicates the optimization of η and ζ to achieve the highest zT . Not only can it be applied into the typical thermoelectric material, it also can be used into the newly reported van der Waals crystal Ta_4SiTe_4 , which exhibits a narrow band gap of $\sim 0.08\text{eV}$. At low temperature, such as 50 K, it is reasonable to assume there is only one kind of carrier in Ta_4SiTe_4 . Consequently, the weighted mobility of electron and chemical potential can be derived. By assuming the linear temperature dependence of chemical potential, the remaining parameters could be obtained. The B_{Kane}^* is calculated to be 0.24 for Ta_4SiTe_4 at 300 K as shown in Figure S13c. The experiment zT is 0.18, which could be optimized to the highest zT of 0.27 shown by the red arrow.

Table S5. Physical parameters for the estimation of B_{Kane}^* in typical $(\text{Bi,Sb})_2\text{Te}_3$ and van der Waals crystal $\text{Ta}_4(\text{Si,P})\text{Te}_4$

Sample	S ($\mu\text{V/K}$)	η	σ (S/cm)	σ_{E0} (S/cm)	κ_L (W/mK)	ζ	B_{Kane}^*
$\text{Bi}_{0.4}\text{Sb}_{1.6}\text{Te}_3$	268	-0.9	402	1207	0.31	4.4	6
$\text{Ta}_4\text{Si}_{0.995}\text{P}_{0.005}\text{Te}_4$	168	0.7	216	281	0.82	3.1	0.24

The following discussion is added into the main text (Page 21).

This elaborately developed B_{Kane}^* not only plays significant role in evaluating and predicting thermoelectric performance in $\text{Pb}_7\text{Bi}_4\text{Se}_{13}$. It can also be extended to thermoelectric materials with narrow band gaps, such as prototypical $(\text{Bi,Sb})_2\text{Te}_3$ and van der Waals crystal Ta_4SiTe_4 . By adopting this B_{Kane}^* , the highest zT of 1.54 and 0.27 are predicted in $(\text{Bi,Sb})_2\text{Te}_3$ and Ta_4SiTe_4 , which could be achieved by furthering optimizing the η and ζ .

4. Finally, the manuscript needs a bit of polishing. For instance, “optimization on” on line 62 should be “optimization of.” The punctuation after Equations 3 and 4 can be improved.

Response: Thanks for this comment. The manuscript has been polished and the punctuations after all equations in the main text and supplementary information have been optimized.

References

1. Ohta M, Chung DY, Kunii M, Kanatzidis MG. Low lattice thermal conductivity in $\text{Pb}_5\text{Bi}_6\text{Se}_{14}$, $\text{Pb}_3\text{Bi}_2\text{S}_6$, and PbBi_2S_4 : promising thermoelectric materials in the cannizzarite, lillianite, and galenobismuthite homologous series. *Journal of Materials Chemistry A* **2**, 20048-20058 (2014).
2. Heinke F, Nietschke F, Fraunhofer C, Dovgaliuk I, Schiller J, Oeckler O. Structure and thermoelectric properties of the silver lead bismuth selenides $\text{Ag}_5\text{Pb}_9\text{Bi}_{19}\text{Se}_{40}$ and $\text{AgPb}_3\text{Bi}_7\text{Se}_{14}$. *Dalton Transactions* **47**, 12431-12438 (2018).
3. Casamento J, *et al.* Crystal structure and thermoelectric properties of the 7,7L Lillianite homologue $\text{Pb}_6\text{Bi}_2\text{Se}_9$. *Inorg Chem* **56**, 261-268 (2017).
4. Li J, *et al.* Thermoelectric Material $\text{SnPb}_2\text{Bi}_2\text{S}_6$: The 4,4L Member of Lillianite Homologous Series with Low Lattice Thermal Conductivity. *Inorg Chem* **58**, 1339-1348 (2018).
5. Sassi S, *et al.* Crystal Structure and Transport Properties of the Homologous Compounds $(\text{PbSe})_5(\text{Bi}_2\text{Se}_3)_3$ ($m=2, 3$). *Inorg Chem* **57**, 422-434 (2018).
6. Lu R, *et al.* High carrier mobility and ultralow thermal conductivity in the synthetic layered superlattice $\text{Sn}_4\text{Bi}_{10}\text{Se}_{19}$. *Materials Advances*, (2021).
7. Heinke F, *et al.* Cornucopia of Structures in the Pseudobinary System $(\text{SnSe})_x\text{Bi}_2\text{Se}_3$: A Crystal-Chemical Copycat. *Inorg Chem* **57**, 4427-4440 (2018).
8. Chen K-B, Lee C-S. Synthesis and characterization of quaternary selenides $\text{Sn}_2\text{Pb}_5\text{Bi}_4\text{Se}_{13}$ and $\text{Sn}_8.65\text{Pb}_0.35\text{Bi}_4\text{Se}_{15}$. *Solid state sciences* **11**, 1666-1672 (2009).
9. Mrozek A, Kanatzidis MG. Tropochemical Cell-Twinning in the New Quaternary Bismuth Selenides $\text{K}_x\text{Sn}_{6-2x}\text{Bi}_{2+x}\text{Se}_9$ and $\text{KSn}_5\text{Bi}_5\text{Se}_{13}$. *Inorg Chem* **42**, 7200-7206 (2003).
10. Agha EC, *et al.* $\text{LiPbSb}_3\text{S}_6$: A semiconducting sulfosalt with very low thermal conductivity. *Inorg Chem* **53**, 673-675 (2014).
11. Shelimova L, Karpinskii O, Konstantinov P, Avilov E, Kretova M, Zemskov V. Crystal Structures and Thermoelectric Properties of Layered Compounds in the $\text{ATe-Bi}_2\text{Te}_3$ (A= Ge, Sn, Pb) Systems. *Inorganic Materials* **40**, 451-460 (2004).
12. Shelimova L, *et al.* Anisotropic thermoelectric properties of the layered compounds PbSb_2Te_4 and PbBi_4Te_7 . *Inorganic Materials* **43**, 125-131 (2007).
13. Zhao J, *et al.* Semiconducting pavonites $\text{CdMBi}_4\text{Se}_8$ (M= Sn and Pb) and their thermoelectric properties. *Chem Mater* **29**, 8494-8503 (2017).

14. Hwang J-Y, Ahn JY, Lee KH, Kim SW. Structural optimization for thermoelectric properties in Cu-Bi-S pavonite compounds. *Journal of Alloys and Compounds* **704**, 282-288 (2017).
15. Zhao J, *et al.* Six quaternary chalcogenides of the pavonite homologous series with ultralow lattice thermal conductivity. *Chem Mater* **31**, 3430-3439 (2019).
16. Khoury JF, *et al.* Quaternary Pavonites $A_{1+x}Sn_{2-x}Bi_{5+x}S_{10}$ (A+= Li+, Na+): Site Occupancy Disorder Defines Electronic Structure. *Inorg Chem* **57**, 2260-2268 (2018).
17. Sassi S, *et al.* Thermoelectric Properties of Polycrystalline n-Type $Pb_5Bi_6Se_{14}$. *Journal of Electronic Materials* **46**, 2790-2796 (2017).
18. Sotnikov AV, Jood P, Ohta M. Enhancing the Thermoelectric Properties of Misfit Layered Sulfides $(MS)_{1.2+q}(NbS_2)_n$ (M= Gd and Dy) through Structural Evolution and Compositional Tuning. *ACS omega* **5**, 13006-13013 (2020).
19. Djieutedjeu H, *et al.* Crystal Structure, Charge Transport, and Magnetic Properties of $MnSb_2Se_4$. Wiley Online Library (2011).
20. Dawahre L, *et al.* Lone-Electron-Pair Micelles Strengthen Bond Anharmonicity in $MnPb_{16}Sb_{14}S_{38}$ Complex Sulfosalt Leading to Ultralow Thermal Conductivity. *ACS Applied Materials & Interfaces* **12**, 44991-44997 (2020).
21. Pei YZ, LaLonde AD, Wang H, Snyder GJ. Low effective mass leading to high thermoelectric performance. *Energ Environ Sci* **5**, 7963-7969 (2012).
22. LaLonde AD, Pei Y, Snyder GJ. Reevaluation of $PbTe_{1-x}I_x$ as high performance n-type thermoelectric material. *Energ Environ Sci* **4**, 2090-2096 (2011).

REVIEWERS' COMMENTS

Reviewer #1 (Remarks to the Author):

The authors have satisfactorily addressed my concerns. Now I can recommend to publish it without revisions.

Reviewer #2 (Remarks to the Author):

In this revised manuscript, the authors have taken into account all the recommendations and criticisms of the reviewers and modified their manuscript accordingly. These modifications have globally improved the readability and conciseness of the manuscript. However, I am still not convinced that the main results are sufficiently novel to deserve publication in Nature Communications. As mentioned by the authors in their rebuttal letter, the convergence of several bands at one specific high-symmetry point of the BZ can favour intervalley scattering, making the convergence of electronic bands not beneficial for the thermoelectric performance. This has been in particular discussed in a recent paper (Park et al., Nature Comm. 12, 3425 (2021)). In the present manuscript, since the authors have not calculated intervalley scattering, the claim that band nesting can be beneficial for thermoelectric performance is not based on a firm theoretical ground.

Furthermore, I am also not convinced by the fact that the quality factor presented in this manuscript is sufficiently novel with respect to prior published results. The authors argue that the novel factor presented takes into account the bipolar effect but TE materials are not operating in this regime due to their reduced performance. Thus, I do not see any significant gain in this formalism compared to what has been done before. The formalism presented is rather an extension of prior results, certainly worth considering for other material systems, but insufficiently relevant to be the key criterion for publication in Nature Comm.

For these reasons, I uphold my previous decision and still believe that the present manuscript would be a better fit for Communication Materials or Scientific Report.

Reviewer #3 (Remarks to the Author):

The authors have extensively improved their paper during the revision. The new mechanical tests in Figures S6, S7, and S8 and the expanded theoretical modelling of Figure S12 have adequately addressed my comments. Furthermore, the extended literature survey and the anisotropic transport measurements in the revised manuscript are quite interesting. I recommend publication to the editor.

Response to the reviewer #1:

Reviewer #1 (Remarks to the Author):

The authors have satisfactorily addressed my concerns. Now I can recommend to publish it without revisions.

Response: Thank you very much for recommending our work to be published in Nature Communications and for your time in reviewing this work.

Response to the reviewer #3:

Reviewer #3 (Remarks to the Author):

The authors have extensively improved their paper during the revision. The new mechanical tests in Figures S6, S7, and S8 and the expanded theoretical modelling of Figure S12 have adequately addressed my comments. Furthermore, the extended literature survey and the anisotropic transport measurements in the revised manuscript are quite interesting. I recommend publication to the editor.

Response: We greatly appreciate the reviewer for recognizing our specific efforts in improving the quality of manuscript and thank his/her recommendation to editors for publication.

Response to the reviewer #2:

Q1: In this revised manuscript, the authors have taken into account all the recommendations and criticisms of the reviewers and modified their manuscript accordingly. These modifications have globally improved the readability and conciseness of the manuscript. However, I am still not convinced that the main results are sufficiently novel to deserve publication in Nature Communications. As mentioned by the authors in their rebuttal letter, the convergence of several bands at one specific high-symmetry point of the BZ can favour intervalley scattering, making the convergence of electronic bands not beneficial for the thermoelectric performance. This has been in particular discussed in a recent paper (Park et al., Nature Comm. 12, 3425 (2021)). In the present manuscript, since the authors have not calculated intervalley scattering, the claim that band nesting can be beneficial for thermoelectric

performance is not based on a firm theoretical ground.

Response: We thank the reviewer for finding the modifications have globally improved the readability and conciseness of the manuscript, and appreciate his/her new comments about the band convergence. First, we would like to point out in our previous response letter and revised manuscript, we only state the point “the convergence of several bands at one specific high-symmetry point of the BZ can induce intervalley scattering”. We did NOT make a conclusion that “the convergence of electronic bands not beneficial for the thermoelectric performance”. In fact, we would like to emphasize that the convergence of nested band is still beneficial for thermoelectric materials, which is not in contrary to the conclusion reported by Park *et al.*¹ Furthermore, the recent work by Park *et al.* supports that thermoelectric materials, such as $\text{Pb}_7\text{Bi}_4\text{Se}_{13}$, holds a great potential for achieving superior thermoelectric performance, by shifting converged bands at same \mathbf{k} point to distant \mathbf{k} points .

Figure 1. A schematic illustration for different configurations of electronic bands. (a) **I**: Only a heavy valence band, VB1, (b) **II**: A heavy valence band, VB1 and a light valence band, VB2 nested at the same \mathbf{k} point in Brillouin Zone (BZ), (c) **III**: VB1 and VB2 nested at the same \mathbf{k} point with a specific offset, (d) **IV**: VB1 and VB2 converged at distant \mathbf{k} points. A schematic illustration for electronic bands in $\text{Pb}_7\text{Bi}_4\text{Se}_{13}$, (e) **V**: Only two heavy conduction bands, CB1 and CB3 converged at Y2 and M2 points. (f) **VI**: two more light conduction bands, CB2 and CB4 nested with CB1 and CB3, respectively. It should be noted that the heavy and light bands are denoted due to their different effective mass. And the adjective underneath Figure (a)-(d) are derived from the conclusion of the recent work.

At first, we will give a brief introduction on the conclusion of the recent work (Park *et al.*, 2021).¹ In the recent work, different configurations of electronic bands have been discussed and compared, as presented in Figure 1a-1d. As shown in Figure 1a and 1b, the band configuration of **I** and **II** correspond to situations of only a heavy

band

Figure 2. Different band configurations and corresponding calculated results in recent work from **Reference [1]** (Park, et al. *Nat. Commun.* 2021, 12 (1), 1-8). (a) Only a heavy valence band (denoted as I), (b) a heavy and a light band nested at Γ point (II), (c) a heavy and a light band nested at Γ point with offset of 0.1 eV (III). (d) Calculated weighted mobility μ_w , and (d) calculated power factors (PFs) for different band arrangements. It is obvious that μ_w and highest PF of band arrangement of III, is higher than that of II and I, which are marked by red arrows in Figure 2d and 2e.

(VB1), and two converged bands (VB1 and VB2). According to theoretical calculation by Park et al,¹ power factors (PFs) of the converged valence bands (VBs) at the same \mathbf{k} point (band nestification) is higher than that with only a heavy band, as marked by red arrows in Figure 2e. Moreover, the weighted mobility (μ_w) of the converged VBs is also higher than that with only a heavy band (red arrows in Figure 2d). According to the calculation results in Figure 2d and 2e, the highest PF and μ_w of the converged valence bands demonstrate considerable increases of 15 % and 24 %, respectively, compared to that with only a heavy band. This demonstrates that the convergence of a heavy and a light band even at the same \mathbf{k} point (band nestification), can still benefit electrical properties. Consequently, the band configuration **II** is better than **I**, as shown in Figure 1a and 1b. Likewise, if there exists a specific band offset

between two nested bands, its PF will be improved further, as shown in Figure 2e. As a result, the **III** shown in Figure 1c can be advanced than that of **I** and **II**. Although the band nestification introduces a certain degree of intervalley scattering, it could increase valley degeneracy, N_v , which is still favorable for the improvement of carrier concentration. The band nestification with offset (**III**, Figure 1c) could not fully comparable with band convergence at distant \mathbf{k} points (**IV**, Figure 1d). However, the critical role of the band nestification with offset in advancing PFs should not be fully negated, especially compared to band configurations of **I** and **II**.

As regard to $\text{Pb}_7\text{Bi}_4\text{Se}_{13}$, a schematic illustration of its electronic bands is presented in Figure 1e and 1f. In Figure 1e, only two heavy conduction bands, CB1 and CB3 work, denoted as band configuration of **V**. This case reflects a simple configuration without band nestification. Furthermore, as shown in Figure 1f, the join of two more light conduction bands, CB2 and CB4, is denoted as band configuration of **VI**. The configuration of **VI** corresponds to the conduction band structure of $\text{Pb}_7\text{Bi}_4\text{Se}_{13}$ as presented in the main text. In **VI**, the heavy CB1 and light CB2 are nested at Y_2 point. Similarly, the heavy CB3 and light CB4 also form band nestification at M_2 point. Finally, the band convergence and band nestification with offsets simultaneously occur among these four conduction bands. According to the above discussion, the **VI** could realize a considerable improvement of thermoelectric efficiency, especially compared to **V** with only two heavy bands, and **I** with one heavy band.

Admittedly, if four bands could be converged at fully distant \mathbf{k} points, it will be a giant promotion for electrical properties in $\text{Pb}_7\text{Bi}_4\text{Se}_{13}$. It should be noted that band convergence at different \mathbf{k} points is commonly found in semiconductors with high crystal symmetries, such as cubic PbCh ($Ch = \text{Te}, \text{Se}$ and S),² cubic GeTe ,³ cubic CoSb_3 ,⁴ *etc.* Table 1 enumerates the typical thermoelectric semiconductors with high valley degeneracy, which demonstrate the significant role of high crystal symmetries. According to group theory, higher group space number corresponds to high symmetric operations. In Table 1, most of typical semiconductors are cubic and their

space group number goes up to 225. The higher crystal symmetry has a close relationship with higher valley degeneracy converged at distant \mathbf{k} points. In sharp contrast, the space group number of $\text{Pb}_7\text{Bi}_4\text{Se}_{13}$ is only 12, which means its symmetry operation number is considerably low. It is rare that such a low-symmetry monoclinic $\text{Pb}_7\text{Bi}_4\text{Se}_{13}$ could also possess reasonable higher valley degeneracy, even some of which are at the same \mathbf{k} point.

In summary, the convergence of electronic bands at distant \mathbf{k} points is superior. However, for semiconductors with lower symmetries, the convergence of bands in distant \mathbf{k} point is a grand challenge to be achieved due to symmetry constrains. Such that, the convergence of nested bands could be a significant alternative for lower-symmetry thermoelectric materials, especially compared to band configuration with only heavy bands. In fact, Pei *et al.* also reported that the band nestification leads to a record high zT in hexagonal Tellurium (*Nat. Commun.* 2016, 7(1), pp.1-6).⁵ In Pei's work, one set of valence bands are nested at H point in BZ. This one set of valence bands include two different valence bands, which finally leads to a high zT of 1.0. In the case of $\text{Pb}_7\text{Bi}_4\text{Se}_{13}$, two sets of bands are nested at M_2 and Y_2 , respectively, which provides multiple charge transport channels and facilitate an advanced thermoelectric performance ($zT = 1.35$). Furthermore, the convergence of nested bands indicates a great potential for thermoelectric materials, if nested bands could be removed away from the same \mathbf{k} point by chemical modifications.

As for intervalley scattering calculation, it is usually performed in small system. Such as CaMg_2Sb_2 in the recent work of Park *et al.*, its primitive cell only includes 5 atoms and the unit cell is only 145 \AA^3 . However, $\text{Pb}_7\text{Bi}_4\text{Se}_{13}$ demonstrates complex structure, with 24 atoms in primitive cell and a unit cell volume of 1383 \AA^3 . The intervalley scattering requires very time-consuming electron-phonon coupling calculations. For such complex and huge system, this calculation is far beyond the ability and scope of present study, especially when our experimental results are strongly supported by theoretical calculations of Park *et al.*

In order to elucidate this issue, the following statement is added into the main

text (Page 12).

A recent study suggests the convergence of electronic bands at distant \mathbf{k} points is superior¹. However, for semiconductors with lower symmetries, this favorable band configuration is a grand challenge. Alternatively, the convergence of nested bands could be significant to advanced electrical properties for lower-symmetry thermoelectric materials. The convergence of nested bands plays an important role in charge transport properties than the situation with only heavy bands, even though it introduces a certain degree of intervalley scattering. This situation is confirmed by the theoretical calculation in the recent work,¹ in which the power factors in band configurations of nested bands are still higher than that in band configuration of heavy band only. Experimentally, the band nestification is proven to be an advanced method in Tellurium thermoelectric compound with space for improvement.⁵ Furthermore, the convergence of nested bands holds a great potential for thermoelectric materials, if nested bands could be removed away from the same \mathbf{k} point by chemical modifications.

Table 1. Crystallographic information of typical thermoelectric materials

Number	Semiconductor	Symmetry	Space group number	Space group
1	$PbCh$ ($Ch = Te, Se, S$) ²	Cubic	225	$Fm\bar{3}m$
2	GeTe ³	Cubic (HT)	225	$Fm\bar{3}m$
		Rhombohedral (LT)	160	R3m
3	Mg_2M ($M=Sn, Si$) ⁶	Cubic	225	$Fm\bar{3}m$
4	SnTe ⁷	Cubic	225	$Fm\bar{3}m$
5	AgSnSbSe ₃ ⁸	Cubic	225	$Fm\bar{3}m$
6	ZrCoBi ⁹	Cubic	216	$F\bar{4}3m$
7	CoSb ₃ ⁴	Cubic	204	$Im\bar{3}$
8	Mg ₃ Sb ₂ ¹⁰	Hexagonal	164	$P\bar{3}m1$
9	BiCuSeO ¹¹	Tetragonal	129	$P4/nmm$
10	Pb ₇ Bi ₄ Se ₁₃	Monoclinic	12	$C2/m$

Q2: Furthermore, I am also not convinced by the fact that the quality factor presented in this manuscript is sufficiently novel with respect to prior published results. The authors argue that the novel factor presented takes into account the bipolar effect but TE materials are not operating in this regime due to their reduced performance. Thus, I do not see any significant gain in this formalism compared to what has been done before. The formalism presented is rather an extension of prior results, certainly worth considering for other material systems, but insufficiently relevant to be the key criterion for publication in Nature Comm.

Response: The reviewer mentioned that TE materials are not operating in the bipolar temperature regime due to their reduced performance, which is inconsistent with some experimental observations^{4, 12}. We disagree with the point with respect. Thermoelectric materials with narrow gaps show bipolar conduction from minority carriers at high temperatures. The bipolar effect affects the contribution of majority carriers. But there indeed exist exceptions that the zT of TE materials still achieve the highest values in this regime where bipolar effect exists.

Figure 3. Bipolar effect in representative thermoelectric material. (a) Seebeck coefficient, and (b) zT values of $\text{Pb}_{1-x}\text{Sb}_{2x/3}\text{Se}$. The red arrows in Figure (a) indicate the maximum of magnitude of Seebeck coefficient due to bipolar effect. The red ellipse indicates highest zT values achieved in the bipolar temperature range. The data of Figure (a) and (b) is from Reference [12] (*Nat. Commun.* 2017, 8(1), 1-8.).

Taking the representative thermoelectric material PbSe as instances, the band gap is approximate 0.28 eV.¹² The bipolar effect is easily found at mediate and high temperature ranges. As shown in Figure 3a, the maximum Seebeck coefficient of $\text{Pb}_{1-x}\text{Sb}_{2x/3}\text{Se}$ ¹² is indicated by red arrows. With one kind of carriers dominating, the Seebeck coefficient should monotonically increase with temperature. Thermally excited minority carriers will cancel out the Seebeck coefficient at mediate and high

temperature ranges, which leads to the decrease of Seebeck coefficient. Figure 3a shows that the bipolar effect plays a significant role above 700 K. However, the zT values for these sample still increase in the bipolar temperature range and finally reach a maximum of ~ 1.6 at 900 K (marked by the red ellipse), due to the interplay of other factors. Thus, this indicates TE materials in the bipolar temperature range still could show enhanced performances, which is of great interest for applications. A lack of an effective approach to evaluate and predict the thermoelectric performance is a great obstacle for the development of narrow-gap thermoelectric materials. Thus, we elaborately develop this new concept of unique quality factor. This unique quality factor is effective to address both band non-parabolicity and bipolar effect, which is challenging for earlier quality factor (B Factor). It is indeed a big step forward for narrow-gap thermoelectric materials.

References

1. Park J, Dylla M, Xia Y, Wood M, Snyder GJ, Jain A. When band convergence is not beneficial for thermoelectrics. *Nat Commun* **12**, 1-8 (2021).
2. Pei YZ, Shi XY, LaLonde A, Wang H, Chen LD, Snyder GJ. Convergence of Electronic Bands for High Performance Bulk Thermoelectrics. *Nature* **473**, 66-69 (2011).
3. Dong JF, *et al.* Medium-temperature thermoelectric GeTe: vacancy suppression and band structure engineering leading to high performance. *Energ Environ Sci* **12**, 1396-1403 (2019).
4. Tang YL, *et al.* Convergence of Multi-Valley Bands as the Electronic Origin of High Thermoelectric Performance in CoSb₃ Skutterudites. *Nat Mater* **14**, 1223-1228 (2015).
5. Lin SQ, Li W, Chen ZW, Shen JW, Ge BH, Pei YZ. Tellurium as a high-performance elemental thermoelectric. *Nat Commun* **7**, (2016).
6. Liu WS, *et al.* n-type thermoelectric material Mg₂Sn_{0.75}Ge_{0.25} for high power generation. *P Natl Acad Sci USA* **112**, 3269-3274 (2015).
7. Tan GJ, *et al.* High Thermoelectric Performance of p-Type SnTe via a Synergistic Band Engineering and Nanostructuring Approach. *J Am Chem Soc* **136**, 7006-7017

(2014).

8. Luo YB, *et al.* High Thermoelectric Performance in the New Cubic Semiconductor AgSnSbSe₃ by High-Entropy Engineering. *J Am Chem Soc* **142**, 15187-15198 (2020).
9. Zhu HT, *et al.* Discovery of ZrCoBi based half Heuslers with high thermoelectric conversion efficiency. *Nat Commun* **9**, (2018).
10. Tamaki H, Sato HK, Kanno T. Isotropic Conduction Network and Defect Chemistry in Mg_{3+δ}Sb₂-Based Layered Zintl Compounds with High Thermoelectric Performance. *Adv Mater* **28**, 10182-10187 (2016).
11. Zhao LD, *et al.* BiCuSeO oxyselenides: new promising thermoelectric materials. *Energ Environ Sci* **7**, 2900-2924 (2014).
12. Chen ZW, *et al.* Vacancy-induced dislocations within grains for high-performance PbSe thermoelectrics. *Nat Commun* **8**, (2017).